# Stochastic Order Learning: An Approach to Rank Estimation Using Noisy Data

## Abstract

A novel algorithm, called stochastic order learning (SOL), for reliable rank estimation in the presence of label noise is proposed in this paper. For noise-robust rank estimation, we first represent label errors as random variables. We then formulate a desideratum that encourages reducing the dissimilarity of an instance from its stochastically related centroids. Based on this desideratum, we develop two loss functions: discriminative loss and stochastic order loss. Employing these two losses, we train a network to construct an embedding space in which instances are arranged according to their ranks. Also, after teaching the network, we identify outliers likely to have extreme label errors and relabel them for data refinement. Extensive experiments on various datasets show that the proposed SOL algorithm yields decent rank estimation results even when labels are corrupted by noise.

## 1 Introduction

Rank estimation — a task to predict the rank or 'ordered class' of an object — is a fundamental problem in machine learning. It has a variety of applications, including facial age estimation (Ricanek & Tesafaye, 2006; Shin et al., 2022), aesthetic score regression (Kong et al., 2016), and medical assessment (Halabi et al., 2019). In many real-world scenarios, however, it is quite challenging to obtain error-free annotations of 'ordered data', as the distinction between adjacent labels is often unclear. For example, in facial age estimation, changes in facial appearance are not visibly apparent over a short age gap. Hence, annotation errors are unavoidable when age labels are collected by human annotators; it was shown by Escalera et al. (2015) that the distribution of apparent ages is different from that of real ages. Label noise also occurs due to the subjectiveness of a labeling task. For aesthetic score regression, there is no universal scoring mechanism, as people have different tastes in beauty and art. Such a subjective nature of aesthetic criteria may lead to unreliable annotations. Variability in labeling is also reported in medical image analysis (Halabi et al., 2019). Thus, to improve reliability, annotations are obtained by averaging the estimates of multiple experts.

Many algorithms have been developed to train machines using imperfect data with noisy labels, but most of them are for classification (Tanno et al., 2019; Song et al., 2019; Ma et al., 2020; Yao et al., 2022; Ye et al., 2023) or segmentation (Yang et al., 2020; Li et al., 2023). Unlike classification or segmentation, rank estimation suffers from varying degrees of label errors due to the ordinal property of classes. Figure 1 compares nominal data for classification and ordered data for rank estimation. In classification, misclassifying a dog as a cat is as harmful as misclassifying a dog as a bear. In contrast, in rank estimation, the error of estimating a 43-year-old as a 59-year-old is severer than that of mistaking a 24-year-old as a 26-year-old. Since noise-robust classification methods treat all noise identically, they are prone to making big estimation errors and are incapable of identifying extreme outliers when applied to ordered data.

Although several noise-robust regression methods exist, regression-based models are known to underperform compared to classification- or ranking-based methods. As pointed out by Zhang et al. (2023), direct regression may fail to learn high-entropy feature representations, resulting in lower mutual information between learned representations and target outputs. Order learning approaches (Lim et al., 2020; Shin et al., 2022; Lee et al., 2022) overcome the limitations of direct regression and have shown promising results in rank estimation. However, these methods assume clean annotations, and their performance degrades in the presence of label noise, highlighting the need for noise-robust order learning algorithms.

(a) Label noise in classification       (b) Label noise in rank estimation

Figure 1: Nominal data in classification versus ordered data in rank estimation. Unlike classification, in rank estimation, certain errors are severer than others.

In this paper, we propose a novel algorithm, stochastic order learning (SOL), to estimate ranks reliably in the presence of label noise. Given a training dataset with noisy labels, we first model the label errors with random variables. Hence, each instance relates stochastically to multiple ranks rather than deterministically to a single rank. We then train an embedding network based on a desideratum, which encourages minimizing stochastic dissimilarities of instances from their corresponding centroids. To achieve this, we design the discriminative loss and the stochastic order loss. Moreover, after the training, we identify outliers, which are likely to have extreme label errors, and relabel them to refine the noisy dataset. Extensive experiments demonstrate that the proposed SOL provides reliable rank estimation results on various ordered datasets. Also, SOL even reduces the overall label noise of a given dataset based on the outlier detection and relabeling.

The contributions of this paper can be summarized as follows.

- We extend the concept of order learning to cope with noisy data by designing a stochastic approach; we model label errors as random variables and derive embedding space constraints to sort instances according to their stochastically related ranks.
- We also propose outlier detection and relabeling schemes to identify instances with extreme label errors and reduce the overall noise level of a given dataset.
- Experiments on various benchmark datasets for facial age estimation, aesthetic score regression, medical image assessment, and textual regression validate the effectiveness of the proposed SOL under label noise.

## 2 RELATED WORK

**Learning from noisy labels:** With the availability of substantial training data, deep learning has shown impressive performance in numerous tasks, but the performance may degrade severely when there is label noise. Thus, learning from noisy labels has been an active area of research; various attempts have been made to alleviate the adverse impacts of label noise. Some are based on robust loss functions (Ghosh et al., 2017; Zhang & Sabuncu, 2018; Lyu & Tsang, 2019; Ma et al., 2020; Ye et al., 2023), or noise-tolerant objectives such as peer loss (Liu & Guo, 2020) that avoid relying on explicit noise-rate estimation. Others include regularization (Tanno et al., 2019; Menon et al., 2020; Xia et al., 2020), robust network architecture (Han et al., 2018a; Goldberger & Ben-Reuven, 2022), selective data sampling (Han et al., 2018b; Jiang et al., 2018; Song et al., 2019), and representation-learning approaches such as selective-supervised contrastive learning (Li et al., 2022). However, these methods focus on classification or segmentation (*i.e.* pixelwise classification) tasks.

Compared to classification, only a few noise-robust regression methods have been developed. Garg & Manwani (2020) first considered label noise in ordinal regression. They, inspired by Natarajan et al. (2013), proposed an unbiased estimator and modified a loss function so that minimizing the modified loss with corrupted labels leads to the same result as minimizing the original loss with clean labels. Castells et al. (2020) down-weighted the contributions of samples with large losses during training, assuming that noisy samples tend to cause large losses. Yao et al. (2022) developed a variant of Mixup (Zhang et al., 2018), which trains on virtual examples interpolated from two training samples. To make Mixup more suitable for regression tasks, they sampled a pair with closer ordinal labels with a higher probability. Wang et al. (2022b) showed that standard regularization schemes are ineffective under label noise, and proposed a noise-robust text regression algorithm that mitigates noise by discarding or repairing detected noisy samples. More recently, Kim et al. (2024) introduced a contrastive fragmentation strategy that partitions the label space into fragments, forms

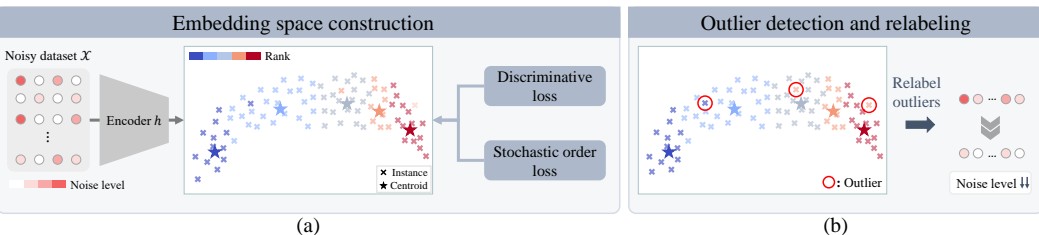

Figure 2: Overview of the proposed SOL algorithm

contrasting fragment pairs, and trains expert extractors on each pair for robust feature learning. They also leveraged neighborhood agreement among the experts to detect clean samples.

**Rank estimation:** Different from ordinary classification, rank estimation aims to predict the ordered class of an object. Early methods estimate object ranks directly using regressors or classifiers. Direct regression (Guo et al., 2009), which predicts scalar values directly, suffers from poor performance in general because it disregards the physical processes underlying ranks, such as aging processes. Classification-based methods (Geng et al., 2007) treat rank estimation as a multi-class classification problem, but they fail to consider the strong ordinal relationship of rank labels. To exploit the ordinal relationship, some ordinal regression methods convert a rank estimation problem into a series of simpler binary classification sub-problems (Frank & Hall, 2001; Li & Lin, 2006). Recently, several techniques have been developed to perform deep ordinal regression effectively, including pairwise regularization (Liu et al., 2018), soft labels (Diaz & Marathe, 2019), continuity-aware probabilistic network (Li et al., 2019), and uncertainty-aware regression (Li et al., 2021). Related to ambiguity modeling, Gao et al. (2017) converted each rank label into a smoothed Gaussian distribution to capture deterministic label uncertainty, but their formulation does not address stochastic label errors.

**Order learning:** Order learning (Lim et al., 2020) is a new approach to rank estimation based on the idea that relative assessment is easier than absolute assessment. Instead of direct prediction, Lim et al. (2020) estimated the rank of an instance by comparing it with references of known ranks. To find more reliable references, Lee & Kim (2021) proposed the order-identity decomposition. Shin et al. (2022) extended the idea of order learning to regression problems, and Lee & Kim (2022) and Lee et al. (2024) developed weakly-supervised and unsupervised techniques for order learning, respectively. Also, Lee et al. (2022) proposed a learning mechanism that exploits not only ordering relations but also metric information among object instances. Similar to the proposed algorithm, they constructed an embedding space in which objects are sorted according to their ranks. However, their algorithm assumes that rank labels are deterministic and error-free, so it fails to model the uncertainty and noise in data. To construct a well-arranged embedding space even in the presence of label noise, we propose a stochastic approach called SOL in this paper.

## 3 PROPOSED ALGORITHM

### 3.1 PROBLEM FORMULATION

There is a training set $\mathcal{X}$, in which each instance is attributed with one of the $n$ ranks (or ordered classes), represented by consecutive integers in $\{1, \ldots, n\}$. Let $\bar{r}_x$ denote the true rank of instance $x \in \mathcal{X}$. However, only a noisy rank $r_x$ is available, given by

$$r_x = \bar{r}_x + e_x \tag{1}$$

where $e_x$ is the label error of $x$. Let $\mathbf{e}$ be the random variable underlying each error $e_x$. It is assumed that $\mathbf{e}$ has a discrete Gaussian distribution;

$$p_s \triangleq \Pr(\mathbf{e} = s) = \frac{1}{C} e^{-\frac{s^2}{2\sigma^2}} \tag{2}$$

where $C = \sum_t e^{-\frac{t^2}{2\sigma^2}}$, and $s, t \in \mathbb{Z}$. Note that the noise distribution in (2) is symmetric ($p_s = p_{-s}$) and unimodal ($p_s \geq p_t$ for $0 \leq s \leq t$). This models label errors in practice. For example, it is more likely for an annotator to mislabel a 10-year-old as 8 or 12 years old than as 20 years old.

We employ an encoder $h$ to map each instance $x \in \mathcal{X}$ into a feature vector $h_x = h(x)$ in an embedding space, as shown in Figure 2. We aim to construct the embedding space in which the

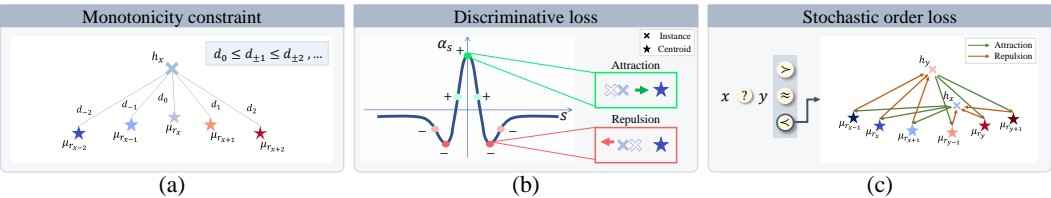

(a)   (b)   (c)

Figure 3: Illustration of the monotonicity constraint and the training losses for constructing a SOL embedding space

instances are arranged according to their ranks, and each 'centroid' $\mu_r$ is the representative vector for instances with rank $r \in \{1, \ldots, n\}$. However, since only the noisy rank $r_x$ in (1) — instead of the true rank $\bar{r}_x$ — is available, instance $x$ relates stochastically to multiple centroids, rather than deterministically to the single centroid $\mu_{\bar{r}_x}$. Specifically, $x$ is associated with $\mu_{r_x-s}$ with probability $p_s$ in (2). Note that, due to the symmetry $p_s = p_{-s}$, $x$ is also associated with $\mu_{r_x+s}$ with $p_s$. Thus, in the embedding space, the mean squared distance $\sum_s p_s d^2(h_x, \mu_{r_x+s})$ should be minimized, where $d$ denotes the Euclidean distance.

We hence define the stochastic dissimilarity of instance $x$ from rank $r$ in the embedding space determined by the encoder $h$ as

$$D_h(x, r) = \sum_s p_s d^2(h_x, \mu_{r+s}). \tag{3}$$

Then, the objective of SOL is to design the encoder $h$ satisfying the following desideratum for each $x \in \mathcal{X}$:

$$D_h(x, r_x) \leq D_h(x, r) \quad \text{for all } r \in \{1, \ldots, n\}. \tag{4}$$

A sufficient condition for satisfying this desideratum is the monotonicity constraint, given by

$$d(h_x, \mu_{r_x+s}) \leq d(h_x, \mu_{r_x+t}) \text{ for all } |s| \leq |t|, \tag{5}$$

as proven in Appendix A. Intuitively speaking, this monotonicity can be achieved, provided that the centroids are arranged directionally according to the ranks, and the instance $h_x$ is located near the centroid $\mu_{r_x}$, as illustrated in Figure 3(a).

In the inference phase, based on the desideratum in (4), we estimate the rank of an unseen instance $x$ by

$$\hat{r}_x = \arg\min_{r \in \{1, \ldots, n\}} D_h(x, r). \tag{6}$$

## 3.2 Stochastic Order Learning

To learn or construct an embedding space in which instances and centroids are well aligned according to the desideratum in (4), we optimize the parameters of the encoder $h$ by minimizing the loss function

$$\ell_{\text{total}} = \sum_{x \in \mathcal{X}} \ell_{\text{disc}}(x) + \sum_{x, y \in \mathcal{X}} \ell_{\text{order}}(x, y) \tag{7}$$

where $\ell_{\text{disc}}$ is the discriminative loss, and $\ell_{\text{order}}$ is the stochastic order loss.

**Discriminative loss:** To encourage the desideratum in (4), we employ the discriminative loss

$$\ell_{\text{disc}}(x) = \sum_{t=1}^{T} \left( D_h(x, r_x) - D_h(x, r_x + t) + D_h(x, r_x) - D_h(x, r_x - t) \right) \tag{8}$$

$$= \sum_{t=1}^{T} \sum_s (2p_s - p_{s-t} - p_{s+t}) d^2(h_x, \mu_{r_x+s}) \tag{9}$$

$$= \sum_s \alpha_s d^2(h_x, \mu_{r_x+s}) \tag{10}$$

where $\alpha_s = \sum_{t=1}^{T} (2p_s - p_{s-t} - p_{s+t})$. Also, $T$ is a hyperparameter, and its impacts are analyzed in Appendix D.1. Note that each term in (8) is non-positive if the desideratum in (4) is satisfied. Thus, minimizing the discriminative loss directly promotes the desideratum.

Also, the coefficient $\alpha_s$ in (10) is a discrete approximation of the 2nd-order derivative of the Gaussian distribution, which has inflection points. Therefore, there exists a threshold $\delta$ such that $\alpha_s$ is positive if $|s| < \delta$, while negative otherwise, as shown in Figure 3(b). Hence, to minimize the discriminative loss, $d(h_x, \mu_{r_x+s})$ should be reduced if $|s| < \delta$. In other words, $h_x$ should be attracted to the centroids for the ranks within the range $(r_x - \delta, r_x + \delta)$. On the contrary, if $|s| > \delta$, $d(h_x, \mu_{r_x+s})$

---

**Algorithm 1** Stochastic Order Learning (SOL)

---

**Input:** A noisy dataset $\mathcal{X}$, $n$ = the number of ranks
 1: Initialize centroids $\{\mu_r\}_{r=1}^n$ via (18)
 2: **repeat**
 3:     Fine-tune the encoder $h$ to minimize $\ell_{\text{total}}$ in (7)            ▷ Network training
 4:     **for all** $r = 1, 2, \ldots, n$ **do**
 5:         Update centroid $\mu_r$ via (18)            ▷ Centroid rule
 6:     **end for**
 7:     **for all** $x \in \mathcal{X}$ **do**
 8:         Estimate the rank of $x$ via (6)
 9:     **end for**
10:     Detect the set of outliers $\bigcup_{r=1}^n \mathcal{X}_r$ via (19)            ▷ Outlier detection
11:     **for all** $x \in \bigcup_{r=1}^n \mathcal{X}_r$ **do**
12:         Estimate the label noise $\hat{e}_x$ via (20)
13:         Refine the label of $x$ via (21)            ▷ Relabeling
14:     **end for**
15: **until** predefined number of epochs
**Output:** Updated labels $\{r_x\}$, centroids $\{\mu_r\}_{r=1}^n$, encoder $h$

---

should be increased, thereby repelling $h_x$ from the centroids for the ranks outside $(r_x - \delta, r_x + \delta)$. To summarize, $\ell_{\text{disc}}$ makes each $h_x$ attracted to the corresponding centroid $\mu_{r_x}$ and its neighbors (to consider the label error), but repelled from the other centroids.

**Stochastic order loss**: In order learning (Lim et al., 2020; Lee & Kim, 2021; Lee et al., 2022), pairwise relationships between instances are used to construct a desired embedding space. Thus, while the discriminative loss $\ell_{\text{disc}}$ in (8) considers the geometric configuration of a single instance $x$ with respect to the centroids, the stochastic order loss $\ell_{\text{order}}$ takes into account the geometric configuration of two instances $x$ and $y$ jointly.

There are three ordering cases between $x$ and $y$ (Lim et al., 2020):

$$x \prec y \text{ if } \bar{r}_x - \bar{r}_y < -\tau, \quad x \approx y \text{ if } |\bar{r}_x - \bar{r}_y| \leq \tau, \quad x \succ y \text{ if } \bar{r}_x - \bar{r}_y > \tau, \tag{11}$$

where $\tau$ is a threshold. For these three cases, Lee et al. (2022) use margin losses to align instances according to the ranks. Similarly, the proposed $\ell_{\text{order}}$ is based on margin losses. But, unlike Lee et al. (2022), true ranks $\bar{r}_x$ and $\bar{r}_y$ are unknown in SOL. Also, each instance relates to multiple centroids randomly in SOL. We hence develop $\ell_{\text{order}}$ to address these differences.

Since only noisy ranks $r_x$ and $r_y$ are available, the true ranks $\bar{r}_x$ and $\bar{r}_y$ in (11) need to be re-represented using (1). Let $s$ and $t$ denote the label noise of samples $x$ and $y$, respectively. Then, $\bar{r}_x - \bar{r}_y = r_x - r_y - s + t$. As we model label noise as stochastic variables, we can compute the probabilities for the three ordering cases using (2):

$$\Pr(x \prec y) = \sum_s \sum_{t:r_x-r_y-s+t<-\tau} p_s p_t, \tag{12}$$

$$\Pr(x \approx y) = \sum_s \sum_{t:|r_x-r_y-s+t|\leq\tau} p_s p_t, \tag{13}$$

$$\Pr(x \succ y) = \sum_s \sum_{t:r_x-r_y-s+t>\tau} p_s p_t. \tag{14}$$

Then, we define the margin loss for the case $x \prec y$ as

$$\ell_{x\prec y} = \sum_{r\leq r_x} \max\{D_h(x,r) - D_h(y,r) + \gamma, 0\} + \sum_{r\geq r_y} \max\{D_h(y,r) - D_h(x,r) + \gamma, 0\} \tag{15}$$

where $\gamma$ is a margin. To minimize the first sum in (15), $D_h(x,r) - D_h(y,r) = \sum_s p_s(d^2(h_x, \mu_{r+s}) - d^2(h_y, \mu_{r+s}))$ should be reduced for $r \leq r_x$. Thus, $h_x$ should be near $\mu_{r+s}$, while $h_y$ should be far from $\mu_{r+s}$. Note that this is enforced for small offsets $s$ only because of the Gaussian weights $p_s$. Similarly, for $r \geq r_y$ and a small $s$, $h_x$ should be far from $\mu_{r+s}$, while $h_y$ should be near $\mu_{r+s}$. Hence, $\ell_{x\prec y}$ helps the arrangement of instances and centroids in the embedding space, as illustrated in Figure 3(c). Note that the loss $\ell_{x\succ y}$ for the case $x \succ y$ is formulated symmetrically.

Also, when $x \approx y$, $h_x$ and $h_y$ should be close to each other. We hence define

$$\ell_{x\approx y} = \sum_{r\in\{1,\ldots,n\}} \max(|D_h(x,r) - D_h(y,r)| - \gamma, 0). \tag{16}$$

Table 1: Performance comparison on the MORPH II dataset.

| | Gaussian | | | | | | Laplacian | | Uniform | | Skewed | |
| | $\kappa = 0.2$ | | $\kappa = 0.3$ | | $\kappa = 0.4$ | | $\kappa = 0.3$ | | $\kappa = 0.3$ | | $\kappa = 0.3$ | |
| Algorithm | MAE(↓) | CS(↑) | MAE(↓) | CS(↑) | MAE(↓) | CS(↑) | MAE(↓) | CS(↑) | MAE(↓) | CS(↑) | MAE(↓) | CS(↑) |
|---|---|---|---|---|---|---|---|---|---|---|---|---|
| SPR (Wang et al., 2022a) | 8.446 | 41.71 | 8.881 | 34.79 | 9.239 | 36.89 | 8.577 | 39.89 | 8.254 | 40.53 | 8.980 | 38.07 |
| ACL (Ye et al., 2023) | 9.017 | 36.75 | 9.492 | 35.61 | 9.314 | 35.74 | 8.873 | 35.87 | 8.849 | 35.95 | 9.613 | 35.93 |
| ROR-CE (Garg & Manwani, 2020) | 2.859 | 86.79 | 3.018 | 86.79 | 3.170 | 82.60 | 3.058 | 84.97 | 2.827 | 87.34 | 3.663 | 77.69 |
| C-Mixup (Yao et al., 2022) | 3.063 | 82.26 | 3.393 | 77.21 | 3.395 | 76.84 | 3.772 | 71.77 | 3.306 | 77.78 | 3.378 | 77.69 |
| ConFrag (Kim et al., 2024) | 2.878 | 84.06 | 3.000 | 82.06 | 3.255 | 78.96 | 3.102 | 80.33 | 2.763 | 84.70 | 3.333 | 78.14 |
| POE (Li et al., 2021) | 2.989 | 82.88 | 3.093 | 80.33 | 3.253 | 79.23 | 3.332 | 77.50 | 2.908 | 83.61 | 3.389 | 75.59 |
| MWR (Shin et al., 2022) | 2.570 | 90.07 | 2.693 | 89.25 | 2.851 | 87.16 | 2.854 | 86.61 | 2.529 | 90.71 | 3.327 | 80.42 |
| GOL (Lee et al., 2022) | 2.516 | 90.89 | 2.671 | 89.07 | 2.861 | 85.97 | 2.846 | 86.16 | 2.509 | 90.26 | 3.351 | 82.51 |
| SOL | **2.489** | **91.35** | **2.663** | **89.62** | **2.826** | **87.70** | **2.794** | **86.89** | **2.499** | **90.89** | **3.296** | **83.15** |

| | Input image | | | | | | Input image | |
|---|---|---|---|---|---|---|---|---|
| | 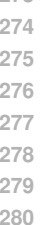 | | | | | | | |
| True label | 17 | 23 | 25 | 42 | 52 | | 23 | 40 |
| SPR | 33 (+16) | 36 (+13) | 43 (+18) | 36 (−6) | 38 (−14) | | 48 (+25) | 25 (−15) |
| GOL | 22 (+5) | 27 (+4) | 20 (−5) | 46 (+4) | 45 (−7) | | 40 (+17) | 29 (−11) |
| SOL | 17 (+0) | 23 (+0) | 25 (+0) | 42 (+0) | 52 (+0) | | 36 (+13) | 33 (−7) |
| | (a) | | | | | | (b) | |

Figure 4: (a) Success and (b) failure cases of age estimation results on the MORPH II dataset. Under each image, we compare the estimated ages of SPR (Wang et al., 2022a), GOL (Lee et al., 2022), and the proposed SOL and specify the corresponding errors inside the parentheses.

Overall, we define the stochastic order loss as

$$\ell_{\text{order}}(x, y) = \Pr(x \succ y)\ell_{x \succ y} + \Pr(x \approx y)\ell_{x \approx y} + \Pr(x \prec y)\ell_{x \prec y}. \quad (17)$$

**Centroid rule**: Moreover, we determine each centroid $\mu_r$ to minimize $\sum_{x \in \mathcal{X}} D_h(x, r_x)$ based on the desideratum in (4),

$$\mu_r = \frac{\sum_{x \in \mathcal{X}} p_{r-r_x} h_x}{\sum_{x \in \mathcal{X}} p_{r-r_x}}, \quad r \in \{1, \dots, n\}, \quad (18)$$

as derived in Appendix B. We update the centroids after every training epoch.

### 3.3 OUTLIER DETECTION AND RELABELING

To obtain a more reliable rank estimator, we identify outliers, likely to have extreme label errors, among instances in the noisy training set and refine their labels by estimating the errors. Then, in turn, we fine-tune the encoder or equivalently revamp the embedding space, so the instances are better arranged based on the refined rank information.

**Outlier detection:** We first estimate the rank of each training instance $x$ using the inference rule in (6). Then, for each rank $r \in \{1, \dots, n\}$, we detect the set $\mathcal{X}_r$ of outliers by

$$\mathcal{X}_r = \{x : r_x = r \text{ and } |r_x - \hat{r}_x| \geq \beta \cdot \max_{y:r_y=r} |r_y - \hat{r}_y|\} \quad (19)$$

where $\beta \in (0, 1)$ is a constant to control the precision of the outlier detection.

**Relabeling:** For each detected outlier $x \in \bigcup_{r=1}^{n} \mathcal{X}_r$, we estimate its label error as

$$\hat{e}_x = \begin{cases} \frac{1}{2|\mathcal{X}|} \sum_{y \in \mathcal{X}} |r_y - \hat{r}_y| & \text{if } r_x > \hat{r}_x, \\ -\frac{1}{2|\mathcal{X}|} \sum_{y \in \mathcal{X}} |r_y - \hat{r}_y| & \text{if } r_x < \hat{r}_x. \end{cases} \quad (20)$$

Then, from (1), we refine the rank of $x$ by

$$r_x \leftarrow r_x - \hat{e}_x. \quad (21)$$

We note that, in (20), $|\hat{e}_x|$ is determined as half of the mean absolute difference between noisy and estimated ranks over all training instances. It is to prevent drastic changes in rank labels, which may rather increase the label errors after relabeling. We repeat the encoder fine-tuning and the outlier detection and relabeling alternately to gradually reduce the label errors and construct a better embedding space. Algorithm 1 summarizes the overall process of SOL.

Table 2: Performance comparison on the CLAP2015 dataset.

| | Gaussian | | | | | | Laplacian | | Uniform | | Skewed | |
| | $\kappa=0.2$ | | $\kappa=0.3$ | | $\kappa=0.4$ | | $\kappa=0.3$ | | $\kappa=0.3$ | | $\kappa=0.3$ | |
| Algorithm | MAE($\downarrow$) | CS($\uparrow$) | MAE($\downarrow$) | CS($\uparrow$) | MAE($\downarrow$) | CS($\uparrow$) | MAE($\downarrow$) | CS($\uparrow$) | MAE($\downarrow$) | CS($\uparrow$) | MAE($\downarrow$) | CS($\uparrow$) |
|---|---|---|---|---|---|---|---|---|---|---|---|---|
| SPR (Wang et al., 2022a) | 9.170 | 44.21 | 9.215 | 43.19 | 9.534 | 40.12 | 9.191 | 38.37 | 9.269 | 43.19 | 9.309 | 45.69 |
| ACL (Ye et al., 2023) | 9.483 | 41.06 | 9.239 | 39.57 | 9.583 | 45.23 | 9.312 | 42.69 | 9.742 | 44.81 | 9.388 | 45.25 |
| ROR-CE (Garg & Manwani, 2020) | 4.163 | 72.85 | 4.432 | 70.06 | 4.900 | 66.27 | 4.789 | 67.19 | 4.174 | 74.42 | 4.650 | 69.42 |
| C-Mixup (Yao et al., 2022) | 5.042 | 61.65 | 5.285 | 58.71 | 5.302 | 58.52 | 4.824 | 62.65 | 4.511 | 64.87 | 4.760 | 63.11 |
| ConFrag (Kim et al., 2024) | 4.898 | 62.19 | 4.658 | 63.11 | 5.328 | 58.20 | 4.690 | 62.47 | 4.858 | 61.17 | 4.512 | 64.97 |
| POE (Li et al., 2021) | 4.052 | 70.34 | 4.169 | 68.86 | 4.390 | 65.52 | 4.303 | 66.64 | 4.061 | 69.32 | 4.401 | 64.97 |
| MWR (Shin et al., 2022) | 3.577 | **79.80** | 3.830 | 76.18 | 4.299 | 72.85 | 4.011 | 74.05 | 3.685 | 77.39 | 4.415 | 70.06 |
| GOL (Lee et al., 2022) | 3.624 | 77.94 | 3.866 | 76.03 | 4.105 | 72.10 | 3.934 | 75.07 | 3.613 | 78.22 | 4.407 | 68.40 |
| SOL | **3.559** | 78.68 | **3.764** | **77.11** | **4.002** | **73.68** | **3.904** | **75.16** | **3.550** | **79.05** | **4.379** | **69.97** |

Table 3: Performance comparison on the AADB dataset.

| | Gaussian | | | | | | Laplacian | | Uniform | | Skewed | |
| | $\kappa=0.2$ | | $\kappa=0.3$ | | $\kappa=0.4$ | | $\kappa=0.3$ | | $\kappa=0.3$ | | $\kappa=0.3$ | |
| Algorithm | MAE($\downarrow$) | CS($\uparrow$) | MAE($\downarrow$) | CS($\uparrow$) | MAE($\downarrow$) | CS($\uparrow$) | MAE($\downarrow$) | CS($\uparrow$) | MAE($\downarrow$) | CS($\uparrow$) | MAE($\downarrow$) | CS($\uparrow$) |
|---|---|---|---|---|---|---|---|---|---|---|---|---|
| SPR (Wang et al., 2022a) | 0.149 | 81.20 | 0.150 | 82.10 | 0.151 | 81.60 | 0.153 | 81.40 | 0.150 | 81.30 | 0.143 | 83.10 |
| ACL (Ye et al., 2023) | 0.147 | 82.90 | 0.148 | 82.50 | 0.157 | 79.43 | 0.151 | 81.50 | 0.153 | 80.80 | 0.153 | 80.74 |
| ROR-CE (Garg & Manwani, 2020) | 0.121 | 88.70 | 0.122 | 89.00 | 0.123 | 88.70 | 0.122 | 89.70 | 0.122 | 90.20 | 0.124 | 89.50 |
| C-Mixup (Yao et al., 2022) | 0.119 | 91.13 | 0.122 | 89.31 | 0.130 | 88.51 | 0.121 | 90.50 | 0.121 | 90.90 | 0.123 | 90.70 |
| ConFrag (Kim et al., 2024) | 0.129 | 88.00 | 0.126 | 88.70 | 0.134 | 86.90 | 0.126 | 89.00 | 0.124 | 89.70 | 0.123 | 88.60 |
| POE (Li et al., 2021) | 0.122 | 89.00 | 0.123 | 89.30 | 0.120 | 89.10 | 0.124 | 89.10 | 0.124 | 88.50 | 0.125 | 88.50 |
| MWR (Shin et al., 2022) | 0.123 | 89.00 | 0.124 | 87.60 | 0.122 | 89.80 | 0.125 | 88.20 | 0.124 | 89.40 | 0.124 | 87.80 |
| GOL (Lee et al., 2022) | 0.114 | 92.40 | 0.117 | 91.80 | 0.119 | 91.00 | 0.118 | 91.50 | 0.117 | 91.60 | 0.120 | 91.00 |
| SOL | **0.111** | **92.70** | **0.114** | **93.20** | **0.115** | **92.00** | **0.115** | **92.30** | **0.116** | **93.30** | **0.118** | **92.30** |

## 4 EXPERIMENTAL RESULTS

We conduct experiments on various datasets for facial age estimation MORPH II (Ricanek & Tesafaye, 2006) and CLAP2015 (Escalera et al., 2015), aesthetic score regression AADB (Kong et al., 2016), medical assessment RSNA (Halabi et al., 2019), and textual regression WMT2020 (Specia et al., 2020). We assess the robustness of the proposed SOL under both synthetic and real-world noisy settings. For synthetic noise, we add Gaussian noise to the rank labels of all training samples, which well reflects real-world noise in ordinal data and is consistent with prior work (Yao et al., 2022; Kim et al., 2024). Specifically, label errors are generated according to the zero-mean discrete Gaussian distribution in (2) with a standard deviation of

$$\sigma = \kappa \cdot \sigma_{\mathcal{X}} \qquad (22)$$

where $\kappa$ is a noise ratio in $(0, 1)$ to control the overall severity of label noise, and $\sigma_{\mathcal{X}}$ is the standard deviation of true rank labels in the training set. In practice, it is unrealistic to know the exact values of $\sigma$ for label errors. Therefore, in the test phase, we use a fixed value of $\sigma_{\text{test}}$ to compute $p_s$ in (2), regardless of $\kappa$. To provide a broader evaluation of robustness, we further consider Laplacian and uniform noise perturbations. For assessment on real-world noisy data, we apply SOL to a textual regression task, where labels are known to be inherently noisy due to subjective human annotations. Additional details of the datasets and noise generation procedures are described in Appendix C.

### 4.1 IMPLEMENTATION

We adopt VGG16 (Simonyan & Zisserman, 2015), initialized with the pre-trained parameters on ILSVRC2012 (Deng et al., 2009), as the encoder $h$. We use the Adam optimizer (Kingma & Ba, 2015) with a batch size of 32 and a weight decay of $5 \times 10^{-4}$. For data augmentation, we do random horizontal flips and random crops. More implementation details including hyperparameter settings are available in Appendix C, and experimental analysis on the hyperparameters is performed in Appendix D.1.

### 4.2 COMPARATIVE ASSESSMENT

We compare the proposed SOL with recent noise-robust classification methods (Wang et al., 2022a; Ye et al., 2023), noise-robust regression methods (Garg & Manwani, 2020; Yao et al., 2022; Kim et al., 2024), and state-of-the-art rank estimators (Li et al., 2021; Shin et al., 2022; Lee et al., 2022). For a fair comparison, the same backbone of VGG16 (Simonyan & Zisserman, 2015) is used for all

Table 4: Performance comparison on the RSNA dataset.

| | Gaussian | | | | | | Laplacian | | Uniform | | Skewed | |
| | $\kappa = 0.1$ | | $\kappa = 0.15$ | | $\kappa = 0.2$ | | $\kappa = 0.15$ | | $\kappa = 0.15$ | | $\kappa = 0.15$ | |
| Algorithm | MAE($\downarrow$) | CS($\uparrow$) | MAE($\downarrow$) | CS($\uparrow$) | MAE($\downarrow$) | CS($\uparrow$) | MAE($\downarrow$) | CS($\uparrow$) | MAE($\downarrow$) | CS($\uparrow$) | MAE($\downarrow$) | CS($\uparrow$) |
|---|---|---|---|---|---|---|---|---|---|---|---|---|
| SPR (Wang et al., 2022a) | 33.80 | 28.50 | 36.48 | 25.00 | 34.88 | 20.50 | 36.77 | 26.50 | 35.50 | 26.00 | 36.85 | 26.50 |
| ACL (Ye et al., 2023) | 35.09 | 26.20 | 35.15 | 26.50 | 35.26 | 25.17 | 33.82 | 24.00 | 34.32 | 22.00 | 35.62 | 20.00 |
| ROR-CE (Garg & Manwani, 2020) | 7.844 | 76.00 | 8.800 | 77.19 | 8.490 | 72.00 | 8.726 | 74.00 | 8.189 | 77.00 | 10.190 | 64.50 |
| C-Mixup (Yao et al., 2022) | 8.200 | 72.40 | 8.621 | 69.71 | 9.054 | 66.70 | 10.603 | 62.00 | 10.124 | 67.00 | 10.504 | 67.00 |
| ConFrag (Kim et al., 2024) | 8.287 | 76.50 | 8.458 | 77.50 | 8.805 | 71.50 | 8.977 | 74.50 | 8.995 | 73.00 | 8.814 | 72.00 |
| POE (Li et al., 2021) | 8.517 | 74.50 | 8.614 | 71.50 | 8.796 | 73.00 | 8.856 | 74.50 | 8.176 | 73.50 | 9.107 | 70.00 |
| MWR (Shin et al., 2022) | 7.833 | 75.00 | 8.239 | 77.50 | 8.353 | 72.00 | 8.272 | 76.00 | 7.939 | 77.50 | 8.741 | 72.50 |
| GOL (Lee et al., 2022) | 8.170 | 77.50 | 7.995 | 80.00 | 8.334 | 75.00 | 8.453 | 72.00 | 7.879 | 77.50 | 8.994 | 71.00 |
| SOL | 7.579 | 78.50 | 7.706 | 80.50 | 8.051 | 76.50 | 8.289 | 76.50 | 7.816 | 78.50 | 8.544 | 73.00 |

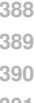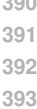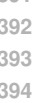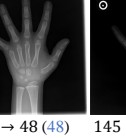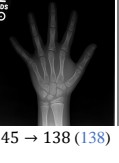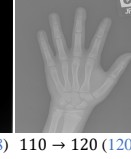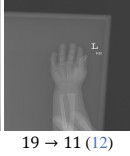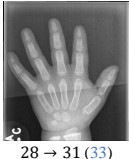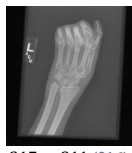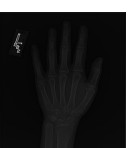

$54 \rightarrow 48\,(48)$  $145 \rightarrow 138\,(138)$  $110 \rightarrow 120\,(120)$  $19 \rightarrow 11\,(12)$  $28 \rightarrow 31\,(33)$     $217 \rightarrow 211\,(216)$  $180 \rightarrow 170\,(189)$

(a)                                                                                     (b)

Figure 5: (a) Success and (b) failure cases of the label refinement on the RSNA training dataset. Under each image, the noisy, refined, and true ranks are specified: noisy $\rightarrow$ refined (true).

methods. For evaluation, we adopt the mean absolute error (MAE) and cumulative score (CS) metrics: MAE is the average absolute error between estimated and ground-truth ranks, and CS computes the percentage of instances whose absolute estimation errors are less than or equal to a tolerance value. The tolerance value is 5 for MORPH II and CLAP2015, 0.25 for AADB, and 12 for RSNA. Justification for the choice of tolerance values is in Appendix C.4.

**Age estimation:** For facial age estimation, we employ two popular datasets MORPH II and CLAP-2015. Table 1 compares the results on MORPH II. SPR (Wang et al., 2022a) and ACL (Ye et al., 2023), which are recent noise-robust classification methods, treat all label errors identically. Compared to rank estimation methods, they underperform because they fail to avoid making large estimation errors (*e.g.* absolute errors bigger than 20). The noise-robust regression methods ROR-CE (Garg & Manwani, 2020), C-Mixup (Yao et al., 2022), and ConFrag (Kim et al., 2024) perform better, for they penalize samples with severe errors. The recent rank estimators MWR (Shin et al., 2022) and GOL (Lee et al., 2022) provide even better results. However, the proposed SOL outperforms all these methods without exception in terms of both MAE and CS.

We also provide examples of age estimation results in Figure 4. These examples are from MORPH II with Gaussian noise at $\kappa = 0.4$. We compare the prediction results on images for which SOL correctly estimates ages in Figure 4(a). Along with the successful cases, we also show some failure cases in Figure 4(b). Note that the noise-robust classifier SPR tends to make big errors as it fails to consider the ordinal property of age labels. The state-of-the-art rank estimator GOL performs better with smaller errors. However, SOL manages to make closer estimates to the true ages than the other algorithms, in both successful and failure cases. Appendix D.12 presents more rank estimation results.

Table 2 lists the performances on CLAP. SOL again achieves the best MAE scores in all settings. Note that GOL also aims to sort instances according to their ranks in an embedding space. Compared to GOL, the proposed SOL provides better results in all cases, and the score gap generally gets bigger as the level of Gaussian noise ($\kappa$) increases. For example, the MAE score gap is 0.103 at $\kappa = 0.4$, while it is 0.065 at $\kappa = 0.2$. These results indicate that, despite label errors, SOL arranges the instances according to their true ranks more reliably. In other words, SOL is more noise-robust than GOL.

**Aesthetic score regression:** Table 3 compares the aesthetic score regression results on AADB. Since aesthetic assessment is inherently subjective and ambiguous, accurately predicting aesthetic scores is highly challenging. Nevertheless, the proposed SOL consistently achieves the best performance across all settings. At the highest Gaussian noise level $\kappa = 0.4$, SOL outperforms the second-best GOL by 3.4% and 1.1% in terms of MAE and CS, respectively. Even at the lowest $\kappa = 0.2$, SOL reduces the MAE by 2.6% and improves the CS by 0.3%.

Table 5: Performance comparison on the WMT2020 dataset

| Algorithm | Real-world noise | |
|---|---|---|
| | PCC($\uparrow$) | SRCC($\uparrow$) |
| Base (Wang et al., 2022b) | 0.645 | 0.612 |
| DIS (Wang et al., 2022b) | 0.653 | 0.627 |
| RES (Wang et al., 2022b) | 0.660 | 0.630 |
| SOL | **0.680** | **0.649** |

Table 6: Ablation studies for the loss functions in (7) on the CLAP2015 dataset.

| Method | $\ell_{\text{disc}}$ | $\ell_{\text{order}}$ | Gaussian | | | | | |
|---|---|---|---|---|---|---|---|---|
| | | | $\kappa = 0.2$ | | $\kappa = 0.3$ | | $\kappa = 0.4$ | |
| | | | MAE($\downarrow$) | CS($\uparrow$) | MAE($\downarrow$) | CS($\uparrow$) | MAE($\downarrow$) | CS($\uparrow$) |
| I | $\checkmark$ | | 20.029 | 14.92 | 16.433 | 20.76 | 18.582 | 17.52 |
| II | | $\checkmark$ | 3.586 | 78.41 | 3.785 | 76.74 | 4.044 | 73.40 |
| III | $\checkmark$ | $\checkmark$ | **3.559** | **78.68** | **3.764** | **77.11** | **4.002** | **73.68** |

**Medical assessment:** In Table 4, we compare the results on the bone age assessment dataset RSNA. The proposed SOL again yields the best results with large margins, with the single exception of the MAE metric for the Laplacian noise. For example, even at $\kappa = 0.1$, SOL outperforms the second-best MWR and GOL with significant gaps of 0.254 and 1.0 in the MAE and CS metrics, respectively. This noise-robustness is meaningful because obtaining error-free annotations on medical datasets is difficult and costly in general.

**Textual regression with real-world noise:** To further validate the effectiveness of SOL, we apply it to a textual regression task in NLP, where labels are known to be noisy due to subjective human annotations. We use the direct assessment (DA) scores from the Ru-En language pairs in WMT2020 (Specia et al., 2020) as regression targets, and follow Wang et al. (2022b) by adopting the same BERT encoder. As shown in Table 5, SOL achieves the best performance with a Pearson's correlation of 0.680 and a Spearman's correlation of 0.649, outperforming the previous state-of-the-art RES by clear margins of 2.0 and 1.9 points, respectively. These results demonstrate that SOL can robustly handle real-world label noise beyond controlled synthetic settings.

**Overall robustness trend:** SOL shows a consistent pattern — its gains over deterministic baselines such as GOL may be modest on relatively clean data, but the advantage steadily grows as noise increases or labels become more subjective.

### 4.3 ANALYSIS

**Label refinement:** SOL refines noisy ranks present in the training dataset using the outlier detection and relabeling scheme in Section 3.3. Figure 5 shows examples of detected outliers in RSNA at $\kappa = 0.15$ (Gaussian). Label errors of up to 10 are well refined in the successful cases in Figure 5(a). In less frequent failure cases, such as Figure 5(b), the refined ranks have bigger errors than the original ones. These are, however, challenging examples because of finger folding or underexposure. More results of the outlier detection and relabeling scheme are provided in Appendices D.4 and D.13.

**Loss functions:** Table 6 compares ablated methods for the loss functions in (7). Method I employs the discriminative loss $\ell_{\text{disc}}$ only, while method II does the stochastic order loss $\ell_{\text{order}}$ only. Compared with method III (SOL), methods I and II degrade the rank estimation results, indicating that both losses contribute to the performance improvement and are complementary to each other. Note that method I yields poor results, for the discriminative loss alone cannot construct a meaningful embedding space; it is trivial to reduce $\ell_{\text{disc}}$ to zero by merging all instances into a single point in the space. However, by comparing II and III, we see that $\ell_{\text{disc}}$ helps to sort instances in the embedding space properly by attracting and repelling instances according to their ranks.

## 5 CONCLUSIONS

The SOL algorithm for rank estimation in the presence of label noise was proposed in this work. First, we represented label errors as random variables. Then, we formulated a desideratum to reduce the dissimilarity of an instance from the stochastically related centroids. Using the discriminative loss and the stochastic order loss, we constructed an embedding space satisfying the desideratum, in which instances are arranged according to their unknown true ranks. Also, we identified outliers, likely to have extreme label errors, and relabelled them for data refinement. Extensive experiments on various rank estimation tasks — including facial age estimation, aesthetic score regression, medical image assessment, and textual regression — demonstrated that SOL yields excellent rank estimation results even when labels are corrupted by noise.

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

## A   DERIVATION OF MONOTONICITY CONSTRAINT IN (5)

The desideratum in (4) can be written as

$$\sum_s p_s d^2(h_x, \mu_{r_x+s}) \leq \sum_s p_s d^2(h_x, \mu_{(r_x+k)+s}) \quad \text{for all } k. \tag{23}$$

For simpler notations, let $L_s \triangleq d^2(h_x, \mu_{r_x+s})$. Then, the desideratum is given by

$$\sum_s p_s L_s \leq \sum_s p_s L_{s+k} \quad \text{for all } k. \tag{24}$$

First, let us consider the case for $k = 1$. From (24), we have

$$\cdots + p_2 L_{-2} + p_1 L_{-1} + p_0 L_0 + p_1 L_1 + p_2 L_2 + \cdots \quad \leq \tag{25}$$
$$\cdots + p_3 L_{-2} + p_2 L_{-1} + p_1 L_0 + p_0 L_1 + p_1 L_2 + \cdots$$

since $p_s$ in (2) is symmetric. Thus,

$$(p_0 - p_1)(L_0 - L_1) + (p_1 - p_2)(L_{-1} - L_2) + (p_2 - p_3)(L_{-2} - L_3) + \cdots \leq 0. \tag{26}$$

Because $p_s$ in (2) is also unimodal, the coefficients $(p_s - p_{s+1})$ are positive for all $s \geq 0$. Hence, the inequality in (26) is satisfied if

$$L_0 \leq L_1, \quad L_{-1} \leq L_2, \quad L_{-2} \leq L_3, \quad \cdots \tag{27}$$

or equivalently

$$L_{-m} \leq L_{1+m} \quad \text{for all } m \geq 0. \tag{28}$$

Next, let us consider the case for $k = 2$. Similar to (26), we have

$$(p_0 - p_2)(L_0 - L_2) + (p_1 - p_3)(L_{-1} - L_3) + (p_2 - p_4)(L_{-2} - L_4) + \cdots \leq 0. \tag{29}$$

This is satisfied if

$$L_{1-m} \leq L_{1+m} \quad \text{for all } m \geq 0. \tag{30}$$

In general, if $k \geq 1$, we have the following condition:

$$L_{\lfloor \frac{k}{2} \rfloor - m} \leq L_{\lceil \frac{k}{2} \rceil + m} \quad \text{for all } m \geq 0. \tag{31}$$

Note that (28) and (30) are special cases of (31). Symmetrically, if $k \leq -1$, we have the condition:

$$L_{\lfloor \frac{k}{2} \rfloor - m} \geq L_{\lceil \frac{k}{2} \rceil + m} \quad \text{for all } m \geq 0. \tag{32}$$

Both conditions in (31) and (32) are satisfied if

$$L_0 \leq L_{\pm 1} \leq L_{\pm 2} \leq L_{\pm 3} \leq \cdots, \tag{33}$$

implying that $L_k$ should be a monotonic increasing function of $|k|$. Rewriting this monotonicity constraint in the original notations, we have the sufficient condition in (5),

$$d(h_x, \mu_{r_x+s}) \leq d(h_x, \mu_{r_x+t}) \quad \text{for all } |s| \leq |t|. \tag{34}$$

## B   DERIVATION OF CENTROID RULE IN (18)

Based on the desideratum in (4), we formulate a cost function

$$J = \sum_{x \in \mathcal{X}} D_h(x, r_x) \tag{35}$$
$$= \sum_{x \in \mathcal{X}} \sum_s p_s d^2(h_x, \mu_{r_x+s}) \tag{36}$$
$$= \sum_{x \in \mathcal{X}} \sum_s p_s(\mu_{r_x+s}^T \mu_{r_x+s} - 2h_x^T \mu_{r_x+s} + h_x^T h_x) \tag{37}$$
$$= \sum_{x \in \mathcal{X}} \sum_r p_{r-r_x}(\mu_r^T \mu_r - 2h_x^T \mu_r + h_x^T h_x). \tag{38}$$

We then update the centroids $\{\mu_r\}_{r=1}^n$ to minimize the cost function $J$. By differentiating $J$ with respect to each $\mu_r$ and setting it to zero, we have

$$\frac{\partial J}{\partial \mu_r} = \sum_{x \in \mathcal{X}} p_{r-r_x}(2\mu_r - 2h_x) = 0. \tag{39}$$

Hence, the optimal centroid is given by

$$\mu_r = \frac{\sum_{x \in \mathcal{X}} p_{r-r_x} h_x}{\sum_{x \in \mathcal{X}} p_{r-r_x}}, \qquad r \in \{1, \ldots, n\}. \tag{40}$$

# C  IMPLEMENTATION DETAILS

## C.1  DATASETS

**MORPH II** (Ricanek & Tesafaye, 2006)**:** It is a dataset for facial age estimation, consisting of 55K facial images in the age range $[16, 77]$. It provides age, gender, and race labels. As in Chang et al. (2011), we use 5,492 Caucasian images divided into training and test sets with a ratio of 8:2.

**CLAP2015** (Escalera et al., 2015)**:** It is for apparent age estimation. The apparent age of each image was rated by at least 10 annotators within the range $[3, 85]$, and the mean rating is used as the ground-truth. This dataset provides 4,691 facial images in total that are split into 2,476 for training, 1,136 for validation, and 1,079 for testing.

**AADB** (Kong et al., 2016)**:** It is a dataset for aesthetic score regression, composed of 10,000 photographs of various themes such as scenery and close-up. We use 8,500 images for training, 500 for validation, and 1,000 for testing. Each image is annotated with an aesthetic score in $[0, 1]$. We quantize the continuous scores with a step size of 0.01 to have 101 discrete ranks.

**RSNA** (Halabi et al., 2019)**:** It is for pediatric bone age assessment, containing 14,236 hand radiographs. We employ the official evaluation protocol in Halabi et al. (2019) — 12,611 for training, 1,425 for validation, and 200 for testing. The bone age range is $[0, 216]$ in months.

**WMT2020** (Specia et al., 2020)**:** It is a dataset for machine translation quality estimation, where translations are scored with human direct assessment (DA) on a scale of $[0, 100]$. The dataset includes seven language pairs of varying resource levels, with sentences mostly sourced from Wikipedia. In this work, we use the Russian→English (Ru-En) subset for evaluation.

## C.2  NOISE DISTRIBUTION SETTINGS

To evaluate the robustness of the proposed SOL, we add random noise generated from three different probability distributions: Gaussian, Laplacian, uniform, and skewed. In all cases, the noise magnitude is controlled by adjusting the noise ratio $\kappa$.

1. Gaussian distribution:
$$\mathbf{e} \sim \mathcal{N}(0, (\kappa \cdot \sigma_\mathcal{X})^2). \tag{41}$$

2. Laplacian distribution:
$$\mathbf{e} \sim \mathrm{Laplace}(0, \kappa \cdot \sigma_\mathcal{X}) \tag{42}$$

    with probability density
$$p(e) = \frac{1}{2\kappa \cdot \sigma_\mathcal{X}} \exp\left(-\frac{|e|}{\kappa \cdot \sigma_\mathcal{X}}\right). \tag{43}$$

3. Uniform distribution:
$$\mathbf{e} \sim \mathcal{U}(-\kappa \cdot \sigma_\mathcal{X}, \, \kappa \cdot \sigma_\mathcal{X}). \tag{44}$$

4. Skewed distribution:
$$\mathbf{e} \sim \mathrm{SkewNorm}(a = 5, \mu = 0, \sigma = \kappa \cdot \sigma_\mathcal{X}). \tag{45}$$

### C.3 Specification of $\sigma$ in (22)

Table 7 specifies the exact values of $\sigma$ for generating the noise in (22) for each dataset.

Table 7: The values of $\sigma$ according to $\kappa$.

|  | $\sigma$ | | | | | |
| --- | --- | --- | --- | --- | --- | --- |
|  | $\kappa = 0.1$ | $\kappa = 0.15$ | $\kappa = 0.2$ | $\kappa = 0.3$ | $\kappa = 0.4$ | $\kappa = 0.5$ |
| MORPH II | 1.092 | 1.638 | 2.184 | 3.276 | 4.368 | 5.460 |
| CLAP2015 | 1.235 | 1.853 | 2.471 | 3.706 | 4.941 | 6.177 |
| AADB | 0.018 | 0.028 | 0.037 | 0.055 | 0.074 | 0.102 |
| RSNA | 4.118 | 6.177 | 8.326 | 12.355 | 16.473 | 20.591 |

### C.4 Tolerance Values for Computing Cumulative Scores

In facial age estimation, the cumulative score (CS) is commonly measured using a tolerance value of 5 (Chang et al., 2011; Shen et al., 2018). For a fair comparison, we also adopt the tolerance value of 5 for the MORPH II and CLAP2015 datasets.

The ranks in AADB, an aesthetic score regression dataset, range from 0 to 1. Thus, for AADB, we use a tolerance value of 0.25, instead of 5.

In medical assessment, previous work only adopts the MAE metric and does not compute CS scores. Bone ages in the RSNA dataset are measured in months instead of years, so RSNA has a bigger error range than facial age estimation datasets. If the same tolerance value 5 is used, it yields very poor CS scores. Thus, we set the tolerance value to be the smallest integer at which the CS scores exceed 75% for all noise ratios $\kappa$. Based on the results in Table 8, we set 12 as the tolerance value for RSNA in all experiments.

Table 8: CS scores (%) of SOL according to the tolerance values on the RSNA dataset (Gaussian label noise).

| Tolerance value | 10 | 11 | **12** | 13 | 14 | 15 | 20 | 25 |
| --- | --- | --- | --- | --- | --- | --- | --- | --- |
| $\kappa = 0.1$ | 71.00 | 75.00 | **78.50** | 82.50 | 84.50 | 87.50 | 94.00 | 97.00 |
| $\kappa = 0.15$ | 68.50 | 74.50 | **80.50** | 85.50 | 86.50 | 89.00 | 95.00 | 97.50 |
| $\kappa = 0.2$ | 69.50 | 73.00 | **76.50** | 80.00 | 84.00 | 86.00 | 92.00 | 99.00 |

We also show the CS curves according to tolerance values on the RSNA dataset in Figure 6. It is observed that the proposed SOL performs better than the state-of-the-art algorithms with the highest area under the curve (AuC) at all noise ratios $\kappa$.

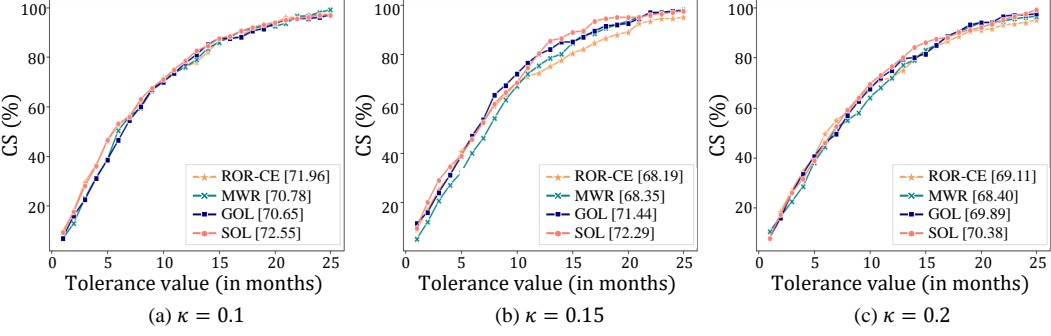

(a) $\kappa = 0.1$      (b) $\kappa = 0.15$      (c) $\kappa = 0.2$

Figure 6: Comparison of the CS curves according to tolerance values on the RSNA dataset (Gaussian label noise). The legend of each graph includes the AuC score for the corresponding algorithm.

## C.5 NETWORK ARCHITECTURE

As described in Section 3.2, we employ an encoder to map each instance into a feature vector in an embedding space. The network structure for the encoder $h$ is specified in Figure 7. The encoder is based on the VGG16 network and takes a $224 \times 224 \times 3$ image as input.

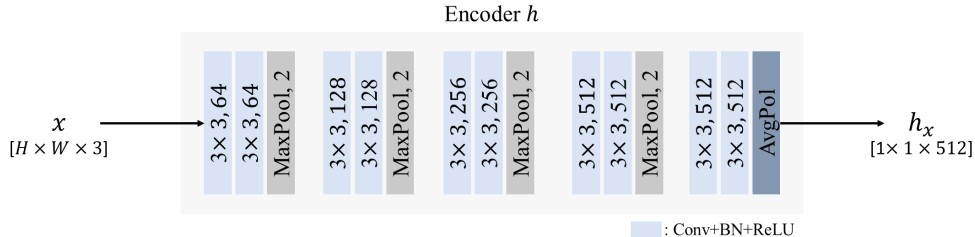

Figure 7: Network structure of the encoder $h$.

## C.6 HYPERPARAMETER SETTINGS

For WMT2020, we train the network for 20 epochs. For all the other datasets, we train the network for 100 epochs. Table 9 summarizes the hyperparameters for each dataset.

Table 9: Hyperparameter settings

| Dataset | Learning rate | Batch size | $T$ in (8) | $\tau$ in (11) | $\gamma$ in (15) | $\beta$ in (19) | $\sigma_{\text{test}}$ |
|---|---|---|---|---|---|---|---|
| MORPH II | $10^{-4}$ | 32 | 1 | 3 | 0.25 | 0.9 | 1 |
| CLAP2015 | $10^{-4}$ | 32 | 1 | 3 | 0.25 | 0.85 | 1 |
| AADB | $5 \times 10^{-5}$ | 32 | 1 | 5 | 0.25 | 0.85 | 0.01 |
| RSNA | $5 \times 10^{-5}$ | 32 | 1 | 3 | 0.25 | 0.9 | 1 |
| WMT2020 | $2 \times 10^{-5}$ | 16 | 1 | 3 | 0.25 | 0.85 | 1 |

# D  MORE EXPERIMENTAL RESULTS

In the following experiments, we use Gaussian distributions for label noise.

## D.1  HYPERPARAMETER ANALYSIS

**Analysis on $T$ in (8):** Table 10 compares the MAE scores at different $T$'s on the CLAP2015 dataset. In this test, $\tau = 3$, $\beta = 0.85$, and $\sigma_{\text{test}} = 1$. Except at $\kappa = 0.2$, where the setting $T = 1$ yields a slightly lower MAE by 0.004 than $T = 3$, the best results are provided by the setting $T = 1$. Thus, we set $T = 1$ as the default mode.

Table 10: MAE scores according to $T$ on the CLAP2015 dataset.

|  | $T = 1$ | $T = 2$ | $T = 3$ |
|---|---|---|---|
| $\kappa = 0.2$ | 3.559 | 3.565 | 3.555 |
| $\kappa = 0.3$ | 3.764 | 3.779 | 3.832 |
| $\kappa = 0.4$ | 4.002 | 4.032 | 4.050 |
| $\kappa = 0.5$ | 4.170 | 4.196 | 4.196 |

**Analysis on $\tau$ in (11):** Table 11 compares the MAE results at different $\tau$'s on CLAP2015. In this test, $T = 1$, $\beta = 0.85$, and $\sigma_{\text{test}} = 1$. Note that $\tau$ is a threshold in (11) to control the balance between rank precision and model robustness. Using $\tau$ as big as 3 achieves robustness and yields decent MAE results. However, when $\tau$ is larger than 3, the performance drops because of the model under-fitting. Hence, we set $\tau = 3$ for CLAP2015.

Table 11: MAE scores according to $\tau$ on the CLAP2015 dataset.

|  | $\tau = 1$ | $\tau = 2$ | $\tau = 3$ | $\tau = 4$ |
|---|---|---|---|---|
| $\kappa = 0.2$ | 3.574 | 3.610 | 3.559 | 3.646 |
| $\kappa = 0.3$ | 3.777 | 3.822 | 3.764 | 3.794 |
| $\kappa = 0.4$ | 4.034 | 3.980 | 4.002 | 4.039 |
| $\kappa = 0.5$ | 4.236 | 4.209 | 4.170 | 4.292 |

**Analysis on $\beta$ in (19):** Table 12 lists the results at different $\beta$'s on CLAP2015. In this test, $T = 1$, $\tau = 3$, and $\sigma_{\text{test}} = 1$. $\beta$ is a parameter to control the precision of outlier detection in (19). Increasing $\beta$ increases the precision, but it also decreases the number of instances that are detected. With a low $\beta$, more instances can be detected as outliers, but there is also the risk of false positives. Generally, the setting $\beta \geq 0.85$ yields better results than $\beta < 0.85$. This is because less precise outlier detection at a low $\beta$ may deteriorate network training by increasing label noise. As specified in Table 9, we set $\beta = 0.85$ for CLAP2015 and AADB and $\beta = 0.9$ for MORPH II and RSNA.

Table 12: MAE scores according to $\beta$ on CLAP2015.

|  | $\beta = 0.8$ | $\beta = 0.85$ | $\beta = 0.9$ | $\beta = 0.95$ |
|---|---|---|---|---|
| $\kappa = 0.2$ | 3.566 | 3.559 | 3.544 | 3.570 |
| $\kappa = 0.3$ | 3.849 | 3.764 | 3.797 | 3.804 |
| $\kappa = 0.4$ | 4.070 | 4.002 | 4.036 | 4.062 |
| $\kappa = 0.5$ | 4.173 | 4.170 | 4.177 | 4.171 |

## D.2 ANALYSIS ON $\sigma_{\text{test}}$

**Gaussian noise assumption and fixed $\sigma_{\text{test}}$:** Many real-world rank-estimation datasets, including CLAP2015 (Escalera et al., 2015), AADB (Kong et al., 2016), and RSNA (Halabi et al., 2019), obtain their ground-truth labels by averaging multiple independent human annotations. Due to the central-limit effect, such averaged labels empirically follow a Gaussian-like distribution; CLAP2015 further provides per-sample variance estimates that directly support this assumption. While individual annotators may deviate from Gaussian behavior, the aggregated labels are typically well approximated by a Gaussian model, making the discrete Gaussian noise formulation in (2) a reasonable choice.

In practice, the true standard deviation of annotation noise is unknown at test time. Therefore, SOL uses a fixed $\sigma_{\text{test}}$ to compute the probabilities $p_s$ in (2). The following analysis evaluates how sensitive SOL is to this hyperparameter.

**Sensitivity to $\sigma_{\text{test}}$:** We examine how the performance of SOL changes with different choices of the fixed $\sigma_{\text{test}}$ used to compute $p_s$ in (2). Table 13 summarizes the MAE results on the CLAP2015 dataset under $T = 1$, $\tau = 3$, and $\beta = 0.85$. A larger $\sigma_{\text{test}}$ couples each instance $x$ more strongly with distant rank centroids, which can weaken rank discrimination. In contrast, a very small value makes the model sensitive to label errors because $x$ interacts only with nearby centroids. Balancing these effects, $\sigma_{\text{test}} = 1.0$ provides the most stable performance in most settings.

Table 13: MAE results according to $\sigma_{\text{test}}$ on the CLAP2015 dataset .

| | $\sigma_{\text{test}} = 0.5$ | $\sigma_{\text{test}} = 1.0$ | $\sigma_{\text{test}} = 1.5$ | $\sigma_{\text{test}} = 2.0$ | $\sigma_{\text{test}} = 2.5$ | $\sigma_{\text{test}} = 3.0$ | $\sigma_{\text{test}} = 3.5$ |
|---|---|---|---|---|---|---|---|
| $\kappa = 0.2$ | 3.555 | 3.559 | 3.548 | 3.549 | 3.588 | 3.593 | 3.670 |
| $\kappa = 0.3$ | 3.801 | 3.764 | 3.794 | 3.797 | 3.848 | 3.888 | 3.985 |
| $\kappa = 0.4$ | 4.000 | 4.002 | 4.072 | 4.070 | 4.061 | 4.194 | 4.355 |
| $\kappa = 0.5$ | 4.198 | 4.170 | 4.203 | 4.288 | 4.259 | 4.343 | 4.499 |

We plot the MAE scores according to $\sigma_{\text{test}}$ in Figure 8. It is observed that MAE results start to degrade significantly once $\sigma_{\text{test}} \geq 4.0$. As shown in Figure 9, the probability distribution $p_s$ in (2) flattens as $\sigma_{\text{test}}$ gets bigger. Thus, the probabilities assigned to different ranks become indistinguishable for SOL to operate well when $\sigma_{\text{test}} \geq 4.0$. Hence, it is appropriate to use a $\sigma_{\text{test}}$ less than 4.0 for CLAP2015.

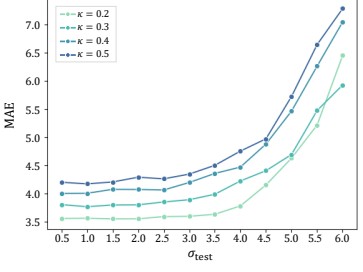

Figure 8: MAE according to $\sigma_{\text{test}}$ on CLAP2015.

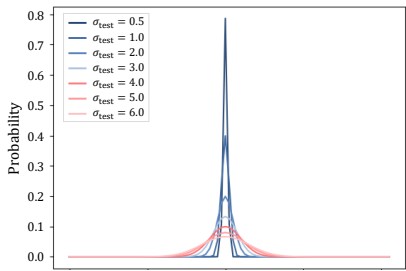

Figure 9: $p_s$ in (2) for different $\sigma_{\text{test}}$.

Table 14 shows a similar trend on the WMT2020 dataset. Although the evaluation metrics differ (PCC and SRCC), the overall variation with respect to $\sigma_{\text{test}}$ remains small, confirming that SOL is not highly sensitive to this hyperparameter in real-world settings. Finally, the $\sigma_{\text{test}}$ values used for all datasets in the main paper are summarized in Table 9.

Table 14: PCC and SRCC scores of SOL on the WMT2020 dataset for different values of $\sigma_{\text{test}}$.

| $\sigma_{test}$ | 0.5 | 1.0 | 1.5 | 2.0 | 2.5 | 3.0 | 3.5 | 4.0 |
|---|---|---|---|---|---|---|---|---|
| PCC (↑) | 0.664 | 0.680 | 0.672 | 0.679 | 0.672 | 0.670 | 0.675 | 0.683 |
| SRCC (↑) | 0.639 | 0.649 | 0.640 | 0.654 | 0.656 | 0.641 | 0.646 | 0.653 |

**Adaptive $\sigma_{\text{test}}$:** To examine whether $\sigma$ can be estimated from data, we add a lightweight head that predicts the mean $\mu$ and standard deviation $\sigma$, trained with a Gaussian negative log-likelihood loss, so that the predicted $\sigma$ replaces the constant in (2). We evaluate two variants: *Joint training*, where the $\sigma$-prediction head and SOL are optimized together, and *Two-stage scheme*, where the $\sigma$-prediction head is trained first and then frozen during SOL training. As shown below for CLAP2015 at $\kappa = 0.4$, the fixed setting achieves better MAE and CS than both adaptive variants.

Table 15: Comparison of adaptive $\sigma_{\text{test}}$ strategies on the CLAP2015 dataset at $\kappa = 0.4$.

| Method | MAE ($\downarrow$) | CS ($\uparrow$) |
|---|---|---|
| Joint adaptive $\sigma_{\text{test}}$ | 5.032 | 67.10 |
| Two-stage adaptive $\sigma_{\text{test}}$ | 4.171 | 71.64 |
| Fixed $\sigma_{\text{test}}$ (default) | **4.002** | **73.68** |

### D.3 LOSS FUNCTIONS

**Alternatives to $\ell_{\text{disc}}$ in (8):** Table 16 compares alternative loss terms for $\ell_{\text{disc}}$. Method I, which is also known as the center loss, aims at directly locating an instance $x$ close to its corresponding centroid $\mu_{r_x}$. On the other hand, method II decreases not only the distance to the corresponding centroid but also to its stochastically-related centroids. Method II performs better than method I. However, the table shows that the proposed discriminative loss $\ell_{\text{disc}}$ yields the best performance.

Table 16: Comparison of alternative choices for $\ell_{\text{disc}}$ in (8) on the CLAP2015 dataset at $\kappa = 0.2$.

| Method | Alternative to $\ell_{\text{disc}}$ | MAE ($\downarrow$) |
|---|---|---|
| I | $d(h_x, \mu_{r_x})$ | 3.593 |
| II | $D_h(x, r_x)$ | 3.585 |
| III | $\ell_{\text{disc}}$ in (8) | 3.559 |

### D.4 OUTLIER DETECTION AND RELABELING

**Impacts of label refinement:** To show the effectiveness of the proposed label refinement (*i.e.* outlier detection and relabeling) scheme, Table 17 compares the results of SOL with and without the label refinement, respectively, on CLAP2015. By examining Table 17 together with Table 2, it can be observed that even without the refinement SOL outperforms the conventional algorithms. However, by applying the refinement scheme, the proposed SOL further improves overall performance. In general, the label refinement reduces label noise in a training dataset, making the training process more reliable. The impact of relabeling also depends on dataset size. Because CLAP2015 is relatively small, only a few samples are identified as outliers, so the quantitative improvements are modest. In contrast, larger datasets such as RSNA contain more inconsistent labels, making the refinement more beneficial. The RSNA results in Table 18 clearly demonstrate this tendency.

Table 17: Comparison of the proposed SOL with and without the label refinement on CLAP2015.

| | $\kappa = 0.2$ | | $\kappa = 0.3$ | | $\kappa = 0.4$ | | $\kappa = 0.5$ | |
|---|---|---|---|---|---|---|---|---|
| Algorithm | MAE ($\downarrow$) | CS ($\uparrow$) | MAE ($\downarrow$) | CS ($\uparrow$) | MAE ($\downarrow$) | CS ($\uparrow$) | MAE ($\downarrow$) | CS ($\uparrow$) |
| w/o label refinement | **3.556** | 78.41 | 3.766 | 76.37 | 4.058 | **73.68** | 4.208 | **72.57** |
| w/ label refinement | 3.559 | **78.68** | **3.764** | **77.11** | **4.002** | 73.68 | **4.170** | 71.64 |

Table 18: Comparison of the proposed SOL with and without the label refinement on RSNA.

| | $\kappa = 0.10$ | | $\kappa = 0.15$ | | $\kappa = 0.20$ | |
|---|---|---|---|---|---|---|
| Algorithm | MAE ($\downarrow$) | CS ($\uparrow$) | MAE ($\downarrow$) | CS ($\uparrow$) | MAE ($\downarrow$) | CS ($\uparrow$) |
| w/o label refinement | 7.967 | **81.50** | 7.800 | 79.50 | 8.196 | 74.00 |
| w/ label refinement | **7.579** | 78.50 | **7.706** | **80.50** | **8.051** | **76.50** |

**Alternative relabeling schemes:** In the proposed relabeling scheme, the ranks of detected outliers are adjusted by the same magnitude via (20). Here, we assess the performance when each detected outlier is relabeled using different magnitudes. Specifically, we adjust the rank of each outlier instance by half of the absolute difference between its noisy and estimated rank. Table 19 lists the results on the CLAP2015 dataset. Compared to method I performing no relabeling, method II improves MAE. However, the proposed relabeling scheme provides the best results. Using the same average value to adjust the ranks prevents drastic changes in rank labels, yielding more reliable performance.

Table 19: Analysis on the relabeling scheme on the CLAP2015 dataset at $\kappa = 0.4$.

|  | Relabeling schemes | MAE ($\downarrow$) | CS ($\uparrow$) |
|---|---|---|---|
| I | No relabeling | 4.058 | **73.68** |
| II | Different magnitudes | 4.012 | 72.75 |
| III | Proposed | **4.002** | 73.68 |

**Noise reduction:** The proposed SOL can refine noisy ranks. To demonstrate this capability, we report MAEs between a noisy rank $r_x$ and the true rank $\bar{r}_x$ and the standard deviations of such noise levels before and after the label refinement in Table 20. In this test, we use the MORPH II and CLAP2015 datasets. Note that the MAE or the standard deviation is reduced in 11 out of 12 tests, confirming the effectiveness of the label refinement. For further analysis, we test how the refinement changes the number of instances at each noise level (*i.e.* label error). Figure 10 plots such statistics on MORPH II at various $\kappa$'s. The red boxes in Figure 10 specify the numbers of instances with high noise levels. We see that the numbers of instances with extreme noise levels are reduced in general. Especially, at $\kappa = 0.4$, the number of instances with $2 \leq e_x \leq 4$ is increased, while that with $e_x \geq 7$ is reduced significantly. It is desirable because severe label errors hinder the construction of a well-sorted embedding space. Consequently, the label refinement generally boosts the performance of SOL.

Table 20: Comparison of the average noise levels before and after the label refinement.

| Noise ratio | MORPH II | | | | | | CLAP2015 | | | | | |
|---|---|---|---|---|---|---|---|---|---|---|---|---|
| | MAE | | | Standard Deviation | | | MAE | | | Standard Deviation | | |
| $\kappa = 0.2$ | 1.737 | $\rightarrow$ | 1.718 | 1.361 | $\rightarrow$ | 1.343 | 1.961 | $\rightarrow$ | 1.959 | 1.508 | $\rightarrow$ | 1.537 |
| $\kappa = 0.3$ | 2.599 | $\rightarrow$ | 2.534 | 1.991 | $\rightarrow$ | 1.942 | 2.970 | $\rightarrow$ | 2.896 | 2.262 | $\rightarrow$ | 2.254 |
| $\kappa = 0.4$ | 3.504 | $\rightarrow$ | 3.401 | 2.638 | $\rightarrow$ | 2.499 | 4.006 | $\rightarrow$ | 3.793 | 3.038 | $\rightarrow$ | 2.899 |

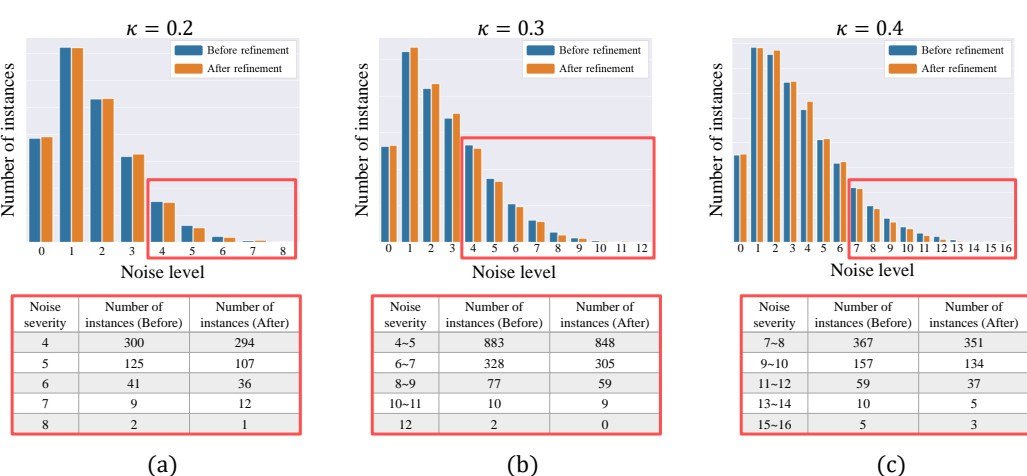

Figure 10: Comparison of the numbers of instances at each noise level before and after the label refinement on the MORPH II dataset.

## D.5 Performance on Partially Noisy Data

In real-world settings, information on which samples are noisy is not given. Hence, for practical use, we assume that all samples have the risk of labeling errors in the experiments in the main paper. However, the proposed SOL is also effective when only a subset of samples are mislabeled. In Table 21, we randomly sample $\varepsilon\%$ of the total dataset and add noise to their labels. The rest of the data is left clean. We compare the proposed SOL to the state-of-the-art algorithm GOL (Lee et al., 2022). In this partially noisy case as well, the proposed SOL generally achieves better performance than GOL.

Table 21: MAE results of GOL / SOL on CLAP2015 when only parts of the total data are corrupted.

|  | $\kappa = 0.2$ | $\kappa = 0.3$ | $\kappa = 0.4$ | $\kappa = 0.5$ |
|---|---|---|---|---|
| $\varepsilon = 10$ | 3.442 / **3.420** | 3.540 / **3.505** | 3.590 / **3.549** | 3.690 / **3.639** |
| $\varepsilon = 20$ | 3.492 / **3.471** | 3.568 / **3.547** | 3.561 / **3.536** | 3.605 / **3.572** |
| $\varepsilon = 30$ | 3.498/ **3.480** | 3.591 / **3.588** | **3.612** / 3.631 | 3.731 / **3.696** |
| $\varepsilon = 40$ | **3.510** / 3.518 | 3.657 / **3.607** | 3.736 / **3.731** | **3.737** / 3.762 |
| $\varepsilon = 50$ | 3.497 / **3.495** | 3.715/ **3.704** | 3.784 / **3.710** | 3.778 / **3.737** |

## D.6 Complexity

**Training time:** Table 22 reports the training time per epoch on the CLAP2015 dataset using an RTX 4090 GPU. We also report the additional runtime introduced by SOL due to its stochastic distance computation and label refinement, by employing GOL as the non-stochastic baseline. While SOL introduces an additional computational cost, it remains practical for training.

Table 22: Training time per epoch on CLAP2015.

| Algorithm | Training time (s) |
|---|---|
| Ranknet | 44.8 |
| SoftRank | 96.2 |
| MWR | 77.3 |
| GOL (non-stochastic) | 27.8 |
| SOL w/o refinement | 39.2 |
| SOL | 52.1 |

We also compare GPU memory usage for loss computation (batch size = 32) in Table 23. GOL consumes substantially more memory, for it constructs full pairwise direction tensors and expanded index structures, which create large intermediate buffers. In contrast, SOL computes pairwise probabilities on the fly without forming dense tensors, resulting in a much smaller memory footprint.

Table 23: GPU memory consumption for loss computation (batch size = 32).

| Algorithm | Memory |
|---|---|
| GOL | 8.19 MB |
| SOL | 0.60 MB |

Table 24 compares the times for computing the centroids in (18) to the total training times. Even for the RSNA dataset consisting of 12,611 training samples, it takes only a few minutes to compute the centroids. This is fast enough for most use cases since the centroids are updated only once per epoch.

Table 24: The processing times (s) required for training one epoch.

|  | MORPH II | CLAP2015 | AADB | RSNA |
|---|---|---|---|---|
| Centroid computation | 6.1 | 5.1 | 39.2 | 286.1 |
| Training 1 epoch | 60.2 | 52.1 | 145.4 | 1160.7 |

**Training speed-up:** Although the centroid computation is not a major bottleneck, its cost can be further reduced by sub-sampling the training instances used during centroid updates. Table 25 reports the MAE performance and the corresponding time complexities for different sampling ratios.

Table 25: Sub-sampling for centroid computation on the CLAP2015 dataset at $\kappa = 0.4$.

| Sampling ratio | MAE | Centroid computation time (s) | Training time per epoch (s) |
|---|---|---|---|
| 0.1 | 4.029 | 0.9 | 47.9 |
| 0.2 | 4.018 | 1.2 | 48.2 |
| 1.0 | 4.002 | 5.1 | 52.1 |

Computing the stochastic distances in FP16 further reduces runtime with negligible impact on MAE, as shown in Table 26.

Table 26: Mixed-precision computation on the CLAP2015 dataset at $\kappa = 0.4$.

| Precision | MAE | Training time per epoch (s) |
|---|---|---|
| FP16 | 4.008 | 48.0 |
| FP32 | 4.002 | 52.1 |

**Training time on RSNA:** Table 27 compares the per-epoch training costs on the RSNA dataset.

Table 27: Training time per epoch on the RSNA dataset.

| Algorithm | Training time per epoch (s) |
|---|---|
| MWR | 1036.3 |
| GOL | 664.1 |
| SOL | 1160.7 |

The large per-epoch cost of SOL on RSNA is due to the data-loading configuration rather than the loss itself. For comparability with prior studies, all methods were evaluated with `num_workers = 1`, which introduces an I/O bottleneck. As shown in Table 28, enabling standard parallel data loading reduces the time from 1160.7 s to 223.6 s. The previously reported 1160.7 s therefore represents a conservative upper bound caused by serial loading; SOL trains efficiently under typical parallel pipelines.

Table 28: Effect of data-loading parallelization on SOL training time for the RSNA dataset.

| num_workers | Training time per epoch (s) |
|---|---|
| 1 | 1160.7 |
| 8 | 223.6 |

**Testing time:** We also compare the average processing time required for testing a single image in Table 29. We use an RTX 4090 GPU and test on the CLAP2015 dataset. For efficiency, we extract the features of all training images and compute the centroids in advance. Therefore, during the test, only the feature extraction of a test image is required. Note that GOL uses $k$-NN while SOL uses the nearest expectation as the inference rule. Compared to GOL, SOL achieves faster inference.

Table 29: The processing times (s) required for testing a single image on CLAP2015.

| Algorithm | Feature extraction (s) | Inference (s) | Total (s) |
|---|---|---|---|
| GOL | 0.040 | 0.083 | 0.123 |
| SOL | 0.040 | 0.051 | 0.091 |

**Memory efficiency:** For large-scale training, memory efficiency is also important. Hence, we compare the number of parameters of SOL with those of conventional methods in Table 30. SOL requires the fewest parameters, indicating its potential for large-scale applications.

Table 30: Comparison of the network complexity.

| Algorithm | # of parameters |
|---|---|
| ACL (Ye et al., 2023) | 134.68M |
| MWR (Shin et al., 2022) | 139.41M |
| GOL (Lee et al., 2022) | 14.75M |
| SOL | 14.72M |

### D.7 INFLUENCE OF LABEL NOISE AT DIFFERENT NOISE RATIOS $\kappa$

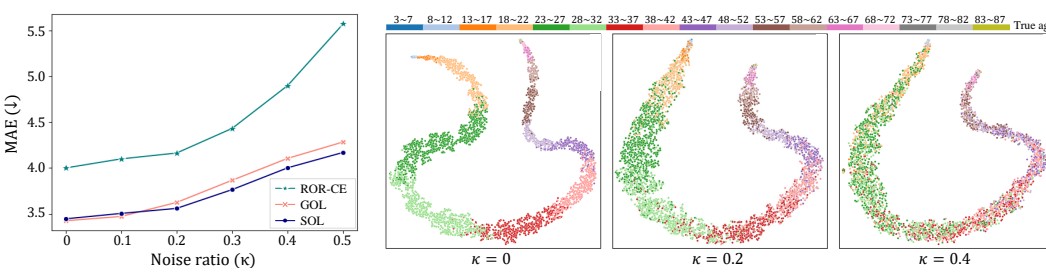

Figure 11: MAE results according to the noise ratio $\kappa$ on CLAP2015.

Figure 12: t-SNE visualization of the embedding spaces for the CLAP2015 dataset at different noise ratios $\kappa$.

**Noise ratios:** Figure 11 analyzes the influence of label noise on the CLAP2015 dataset, by comparing the proposed SOL with ROR-CE and GOL at different noise ratios $\kappa$. For each algorithm, the increase in $\kappa$ degrades the MAE performance. However, the degradation of the conventional algorithms is severer than that of SOL, demonstrating the superior noise-robustness of SOL.

**Embedding spaces:** Figure 12 visualizes the embedding spaces of SOL using t-SNE (Maaten & Hinton, 2008). As $\kappa$ increases, different ages are more mixed up in the space due to bigger label errors. However, at all $\kappa$, the instances are generally well aligned according to their true ages. We show more t-SNE visualizations in Appendix D.11.

### D.8 COMPARISON TO LEARNING-TO-RANK METHODS

For a more complete comparison with learning-to-rank techniques, we additionally implemented RankNet (Burges et al., 2005) and SoftRank (Taylor et al., 2008) under our experimental setup. Both models were trained using the same VGG16 backbone and evaluated through k-NN regression. The results on the MORPH II dataset are summarized in Table 31.

Table 31: Comparison with RankNet and SoftRank on the MORPH II dataset.

| | Gaussian | | | | | | Laplacian | | Uniform | | Skewed | |
|---|---|---|---|---|---|---|---|---|---|---|---|---|
| | $\kappa = 0.2$ | | $\kappa = 0.3$ | | $\kappa = 0.4$ | | $\kappa = 0.3$ | | $\kappa = 0.3$ | | $\kappa = 0.3$ | |
| Algorithm | MAE($\downarrow$) | CS($\uparrow$) | MAE($\downarrow$) | CS($\uparrow$) | MAE($\downarrow$) | CS($\uparrow$) | MAE($\downarrow$) | CS($\uparrow$) | MAE($\downarrow$) | CS($\uparrow$) | MAE($\downarrow$) | CS($\uparrow$) |
| RankNet (Burges et al., 2005) | 2.639 | 89.80 | 2.990 | 86.16 | 3.116 | 82.79 | 3.146 | 84.15 | 2.634 | 88.89 | 3.490 | 80.97 |
| SoftRank (Taylor et al., 2008) | 3.147 | 83.06 | 3.394 | 81.97 | 3.427 | 80.15 | 3.801 | 75.96 | 3.137 | 84.34 | 4.018 | 73.32 |
| SOL | **2.489** | **91.35** | **2.663** | **89.62** | **2.826** | **87.70** | **2.794** | **86.89** | **2.499** | **90.89** | **3.296** | **83.15** |

## D.9 OUTLIERS IN THE WMT2020 DATASET

We provide a qualitative analysis of outlier cases detected by SOL on the real-noise WMT2020 translation-quality dataset. Unlike synthetic noise, discrepancies in WMT2020 originate from genuine human variability, including strong penalties applied to fluent translations and unexpectedly high scores assigned to mistranslated or semantically incorrect outputs. Typical outliers are categorized into two classes.

- Type A: fluent or semantically acceptable translations that receive abnormally low human scores,
- Type B: mistranslated or semantically incorrect outputs that nevertheless receive unusually high scores.

Table 32 presents representative examples identified by SOL. Each case exhibits a clear mismatch between linguistic quality and the annotated score, highlighting the presence of nontrivial and asymmetric annotation noise in WMT2020.

Table 32: Representative outliers detected by SOL on the WMT2020 dataset.

| Type | Real Score | Pred Score | Source Text | Translation | Issue |
|------|-----------|-----------|-------------|-------------|-------|
| A1 | 4 | 22 | Ne po cheloveku spes'. | Don't rush into it. | Fluent sentence but unusually low human score. |
| A2 | 6 | 17 | Ne penyay na zerkalo, kol' rozha kriva. | Don't foam at the mirror if it's crooked. | Acceptable fluency, score is unrealistically low. |
| B1 | 66 | 6 | Zadkom, kuvyrkom, da i pod gorku. | Backward, somersault, and downhill. | Literal mistranslation; idiomatic meaning ("things going downhill") is lost. |
| B2 | 56 | 8 | Religiya yad – beregi rebyat. | Religion Poison – Save the Children | Ungrammatical; missing verb ("Religion is poison"), resulting in awkward phrasing. |
| B3 | 67 | 15 | Chto za chudak, da i chudilo. | What a freak, and a miracle. | Semantic error; "chudilo" mistranslated as "miracle," losing intended meaning. |

## D.10 ABLATION STUDIES AND ANALYSIS ON ADDITIONAL DATASETS

To verify that the same design choices transfer beyond CLAP2015, we conducted ablation studies on RSNA (Gaussian noise with $\kappa = 0.15$) and WMT2020. As summarized in Table 33, both datasets follow the same pattern observed earlier: using either $l_{\text{disc}}$ or $l_{\text{order}}$ alone provides partial performance gains, whereas combining both terms yields the best results.

Table 33: Ablation studies on RSNA and WMT2020.

| Method | $l_{\text{disc}}$ | $l_{\text{order}}$ | RSNA | | WMT2020 | |
|--------|------------------|-------------------|-----------------|-----------------|-----------------|-----------------|
| | | | MAE ($\downarrow$) | CS ($\uparrow$) | PCC ($\uparrow$) | SRCC ($\uparrow$) |
| I | ✓ | | 8.357 | 74.50 | 0.396 | 0.354 |
| II | | ✓ | 8.040 | 77.50 | 0.673 | 0.634 |
| III | ✓ | ✓ | **7.706** | **80.50** | **0.680** | **0.649** |

### D.11 MORE t-SNE VISUALIZATIONS

We visualize the embedding spaces according to different noise ratios $\kappa$ using t-SNE. The t-SNE plots for the MORPH II, AADB, and RSNA datasets are shown in Figures 13, 14, and 15, respectively.

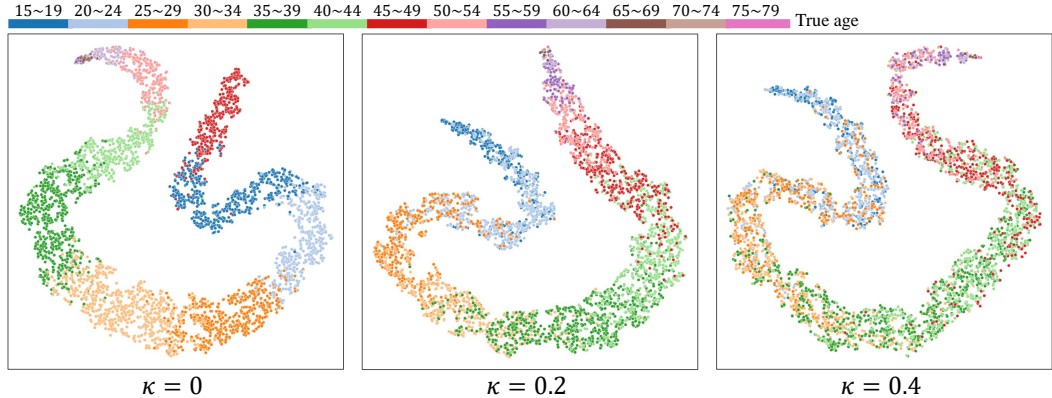

Figure 13: t-SNE visualization of the embedding spaces for MORPH II at different noise ratios $\kappa$.

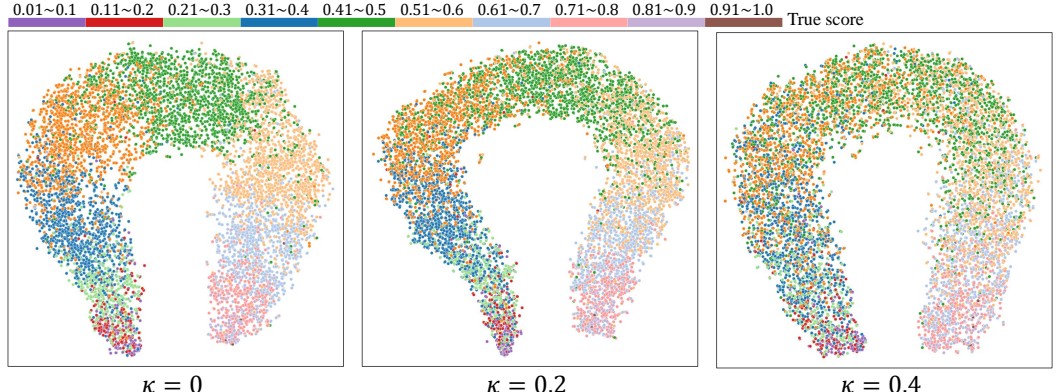

Figure 14: t-SNE visualization of the embedding spaces for AADB at different noise ratios $\kappa$.

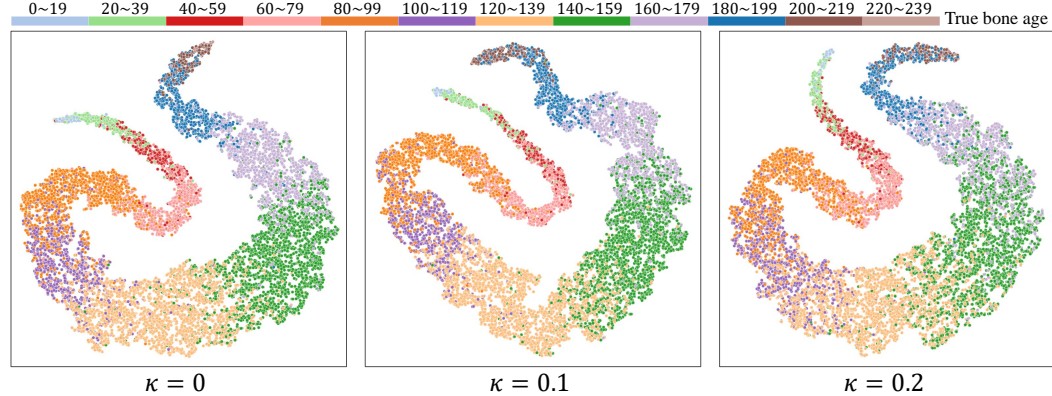

Figure 15: t-SNE visualization of the embedding spaces for RSNA at different noise ratios $\kappa$.

## D.12 MORE RANK ESTIMATION EXAMPLES

Figures 16, 17, and 18 show rank estimation results of the proposed SOL on the CLAP2015, AADB, and RSNA datasets, respectively.

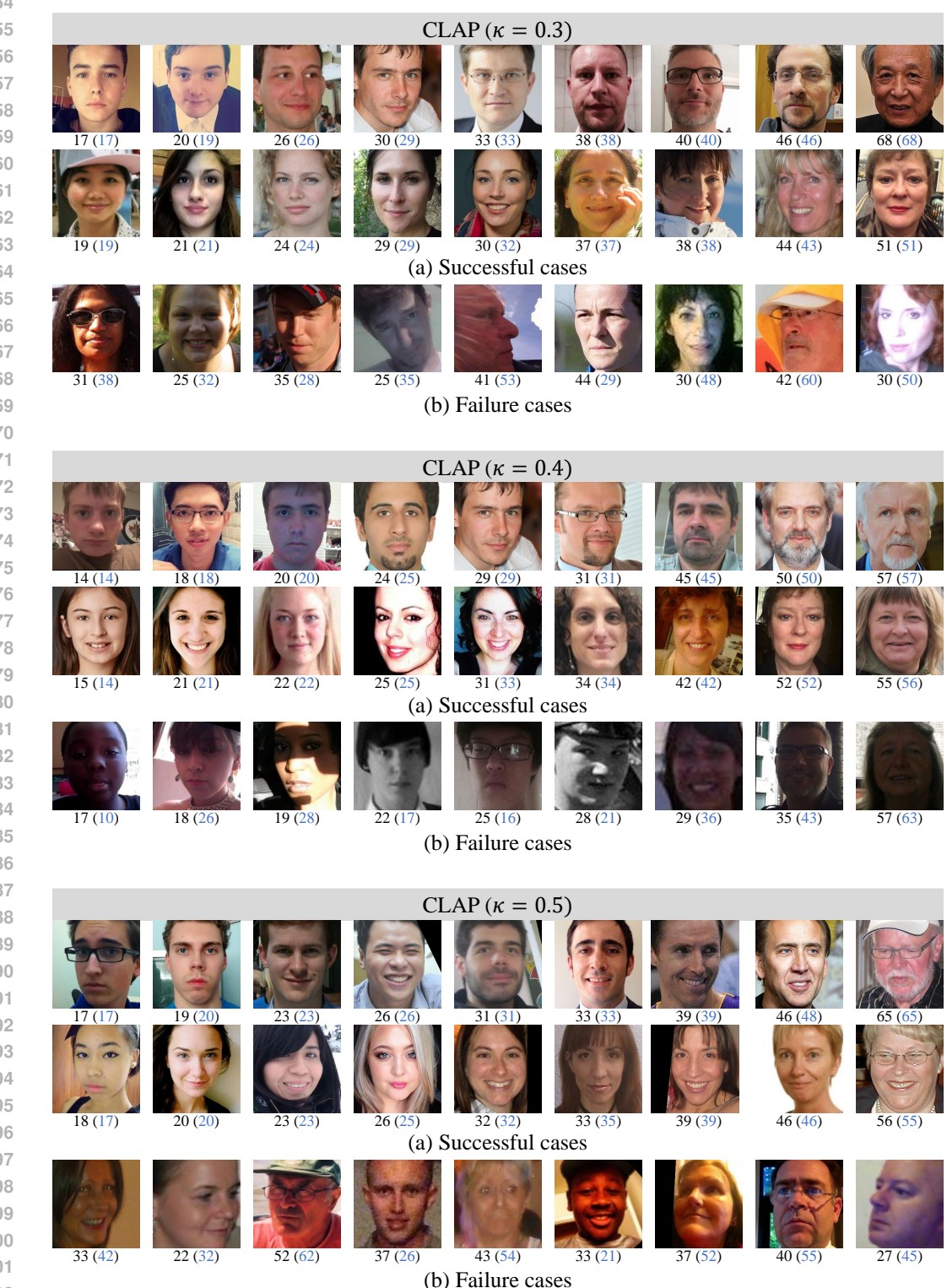

Figure 16: (a) Success and (b) failure cases of age estimation results on the CLAP2015 dataset. Under each image, the estimated ages are specified with the ground-truth in parentheses.

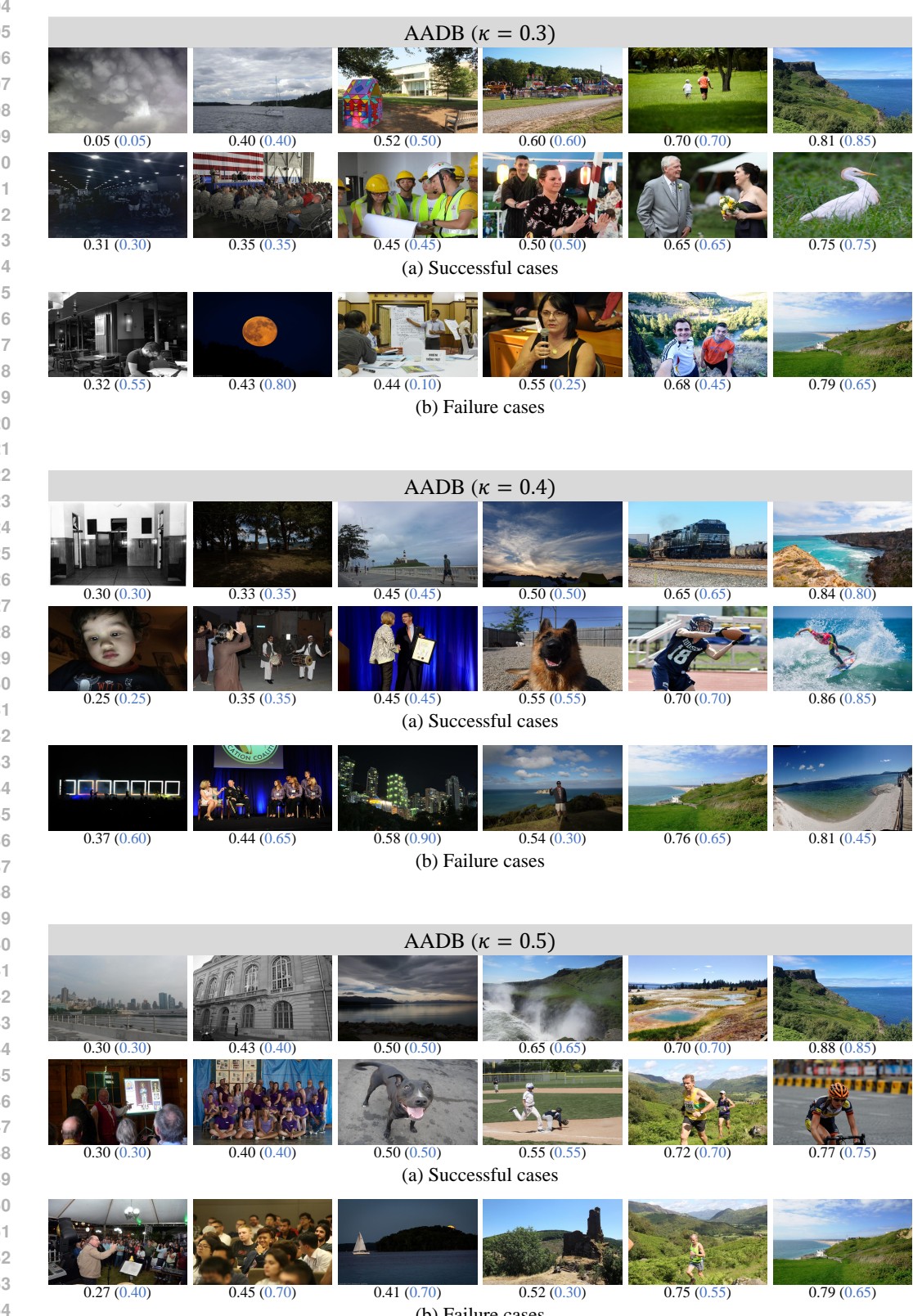

Figure 17: (a) Success and (b) failure cases of aesthetic score estimation results on the AADB dataset. Under each image, the estimated scores are specified with the ground-truth in parentheses.

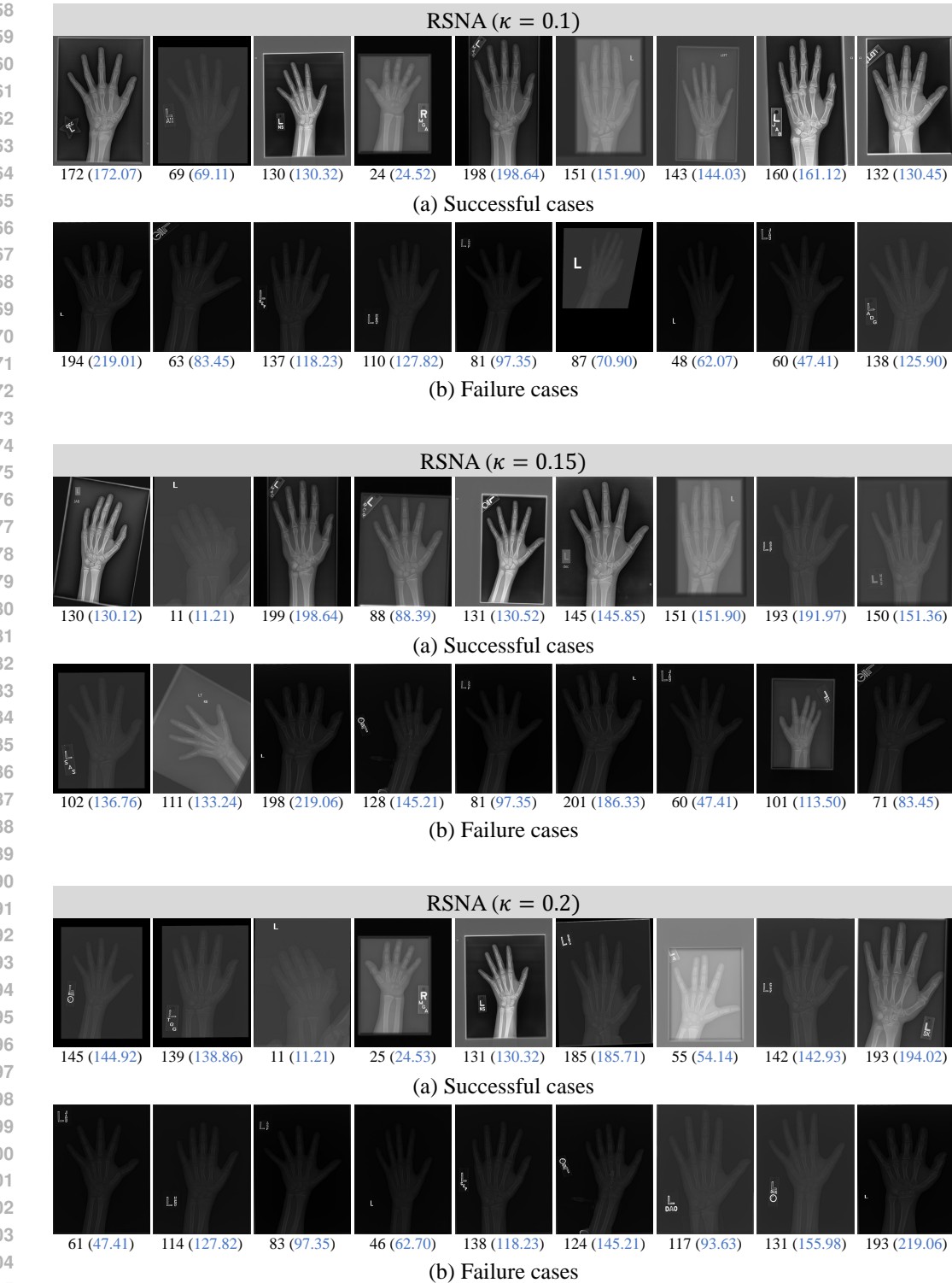

Figure 18: (a) Success and (b) failure cases of bone age assessment results on the RSNA dataset. Under each image, the estimated ages (in months) are specified with the ground-truth in parentheses.

## D.13 MORE EXAMPLES OF DETECTED OUTLIERS

Figures 19, 20, and 21 show examples of detected outliers on the MORPH II, CLAP2015, and AADB datasets, respectively.

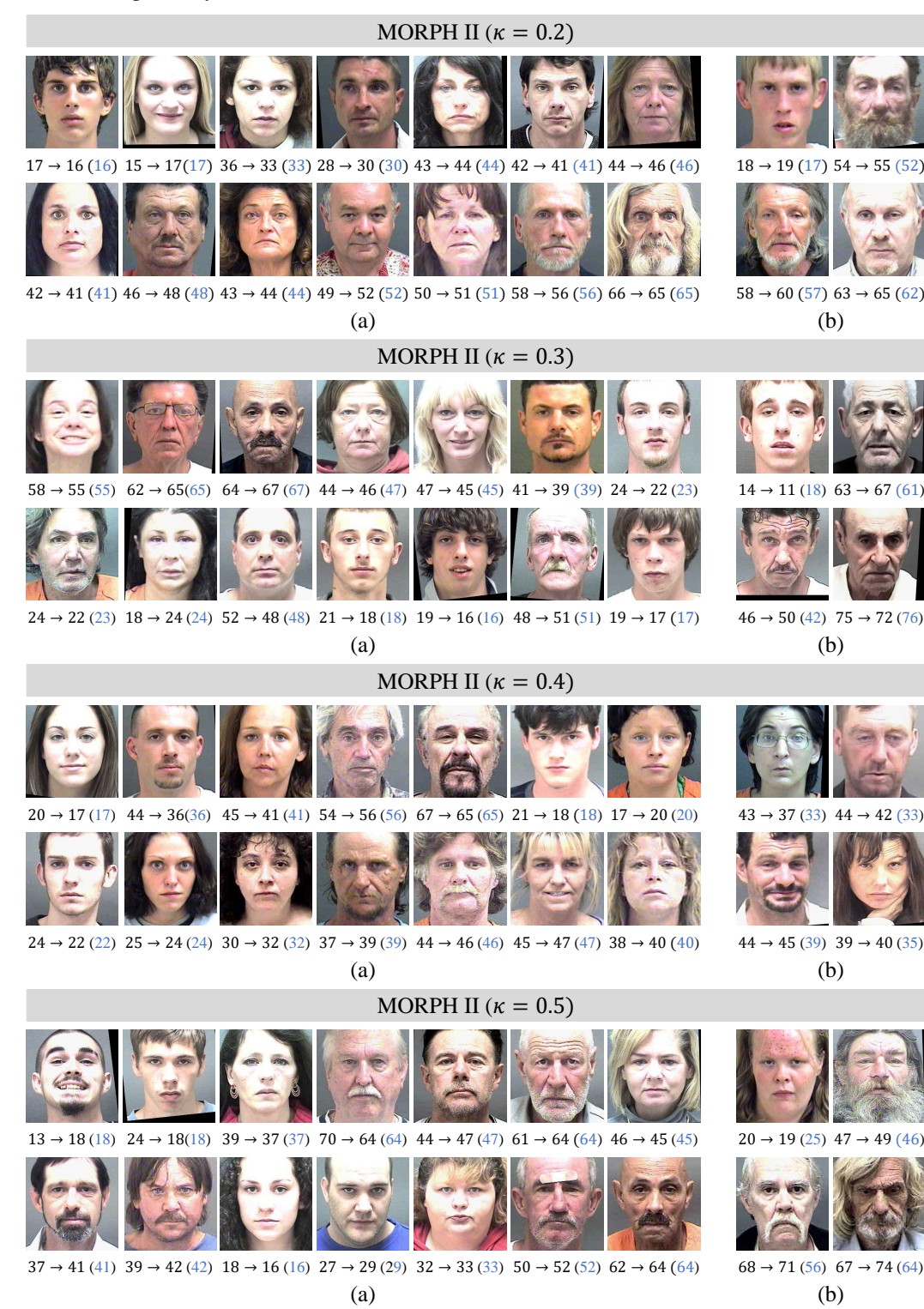

Figure 19: (a) Success and (b) failure cases of the label refinement on the MORPH II dataset. Under each image, the noisy, refined, and true ranks are specified: noisy → refined (true).

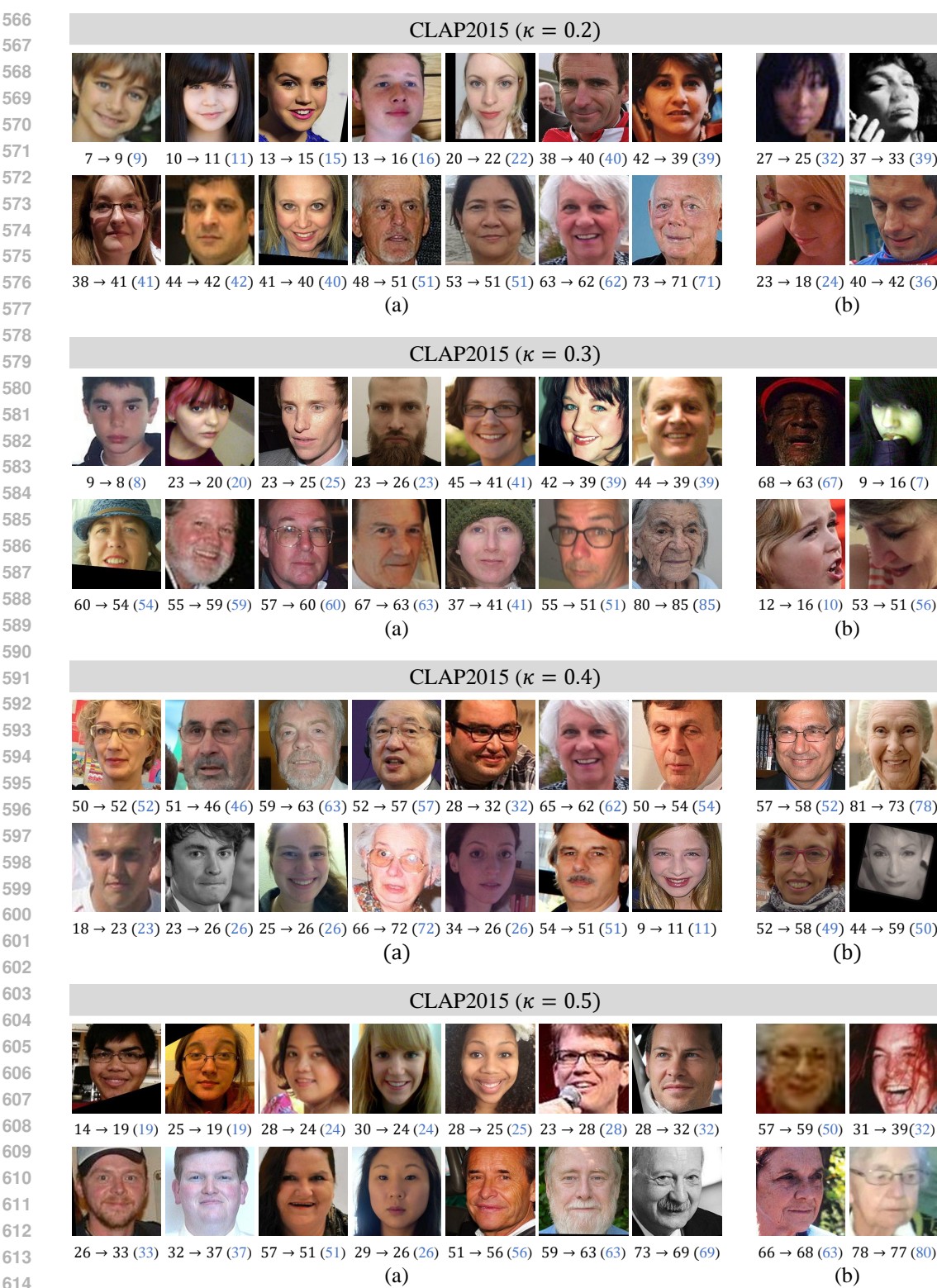

Figure 20: (a) Success and (b) failure cases of the label refinement on the CLAP dataset. Under each image, the noisy, refined, and true ranks are specified: noisy → refined (true).

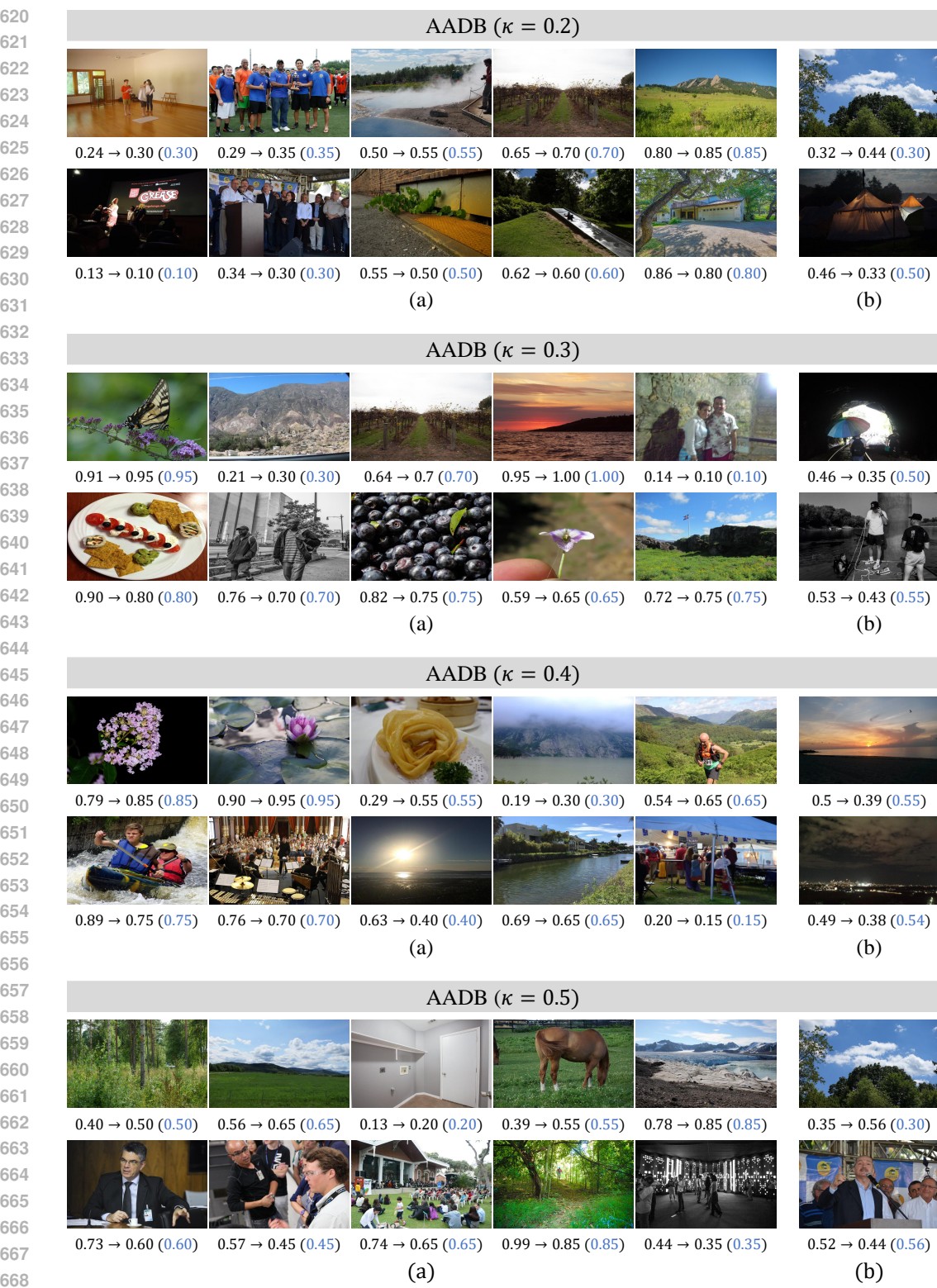

Figure 21: (a) Success and (b) failure cases of the label refinement on the AADB dataset. Under each image, the noisy, refined, and true ranks are specified: noisy → refined (true).

## E   BROADER IMPACTS

Due to the intrinsic imbalance of facial datasets (Ricanek & Tesafaye, 2006; Escalera et al., 2015), there may be unwanted gender or racial bias for deep learning-based facial analysis methods. When trained on such facial datasets, the proposed algorithm is not free from this bias either. Thus, the bias should be resolved before any practical usage. We recommend using the proposed algorithm for research only.

