# OpenReview forum: "Stochastic Order Learning: An Approach to Rank Estimation Using Noisy Data"
_ICLR.cc/2026/Conference — ICLR 2026 Conference Withdrawn Submission_

### Official Review · Reviewer_9juu · 2025-10-25

**Soundness:** 4
**Presentation:** 3
**Contribution:** 3
**Rating:** 6
**Confidence:** 4

**Summary:**

This paper proposes Stochastic Order Learning (SOL), a novel algorithm for robust rank estimation when data labels are corrupted by noise. The key contributions include modeling label errors as random variables, formulating a desideratum based on minimizing stochastic dissimilarity from centroids, and introducing two new loss functions: discriminative loss and stochastic order loss. The method also incorporates an outlier detection and relabeling scheme to refine noisy training data. Extensive experiments on various datasets (facial age estimation, aesthetic score regression, medical assessment, and textual regression) demonstrate that SOL achieves state-of-the-art performance and exhibits strong robustness to label noise.

**Strengths:**

1. The core idea of treating label errors as random variables and formulating the objective based on minimizing stochastic dissimilarity is a theoretically sound approach to handling noise in ordinal data.
2. The paper introduces two complementary loss functions: the discriminative loss and the stochastic order loss. The former is for embedding space construction by attracting to neighboring centroids and repelling from distant ones, and the latter enforces pairwise ordering relationships in a stochastic manner.
3. The outlier detection and relabeling scheme provides a practical way to refine noisy ranks, improving the overall reliability of the training data.
4. The algorithm is tested extensively across a variety of rank estimation tasks, including facial age estimation (MORPH II, CLAP2015), aesthetic score regression (AADB), medical assessment (RSNA), and textual regression (WMT2020), under synthetic (Gaussian, Laplacian, Uniform) and real-world noise settings.

**Weaknesses:**

1. While experiments demonstrate that the label refinement generally reduces MAE and standard deviation of noise, the relabeling scheme (Equation 20) uses a heuristic approach for the magnitude of label error correction (half of the mean absolute difference over all training instances). A stronger theoretical justification or a more adaptive mechanism for this step could enhance its reliability.
2. Some Missing related works[1,2,3]. Expecially, [1] used the normal distribution and the Gaussian kernel to model label ambiguity, which is similar to the noise modeling in this work.

[1] Gao, Bin-Bin, et al. "Deep label distribution learning with label ambiguity." TIP 2017

[2] Li, Shikun, et al. "Selective-supervised contrastive learning with noisy labels." CVPR 2022

[3] Liu, Yang, and Hongyi Guo. "Peer loss functions: Learning from noisy labels without knowing noise rates."  ICML 2020

**Questions:**

See above.

---

> ### Author Response · Authors · 2025-11-17
> **We thank the reviewer for the positive evaluation and constructive feedback.**
>
> Thank you for your positive review and insightful suggestions. Please find our responses below; the corresponding revisions have been incorporated into the paper and highlighted in blue.
>
> ***
> > **Relabeling scheme**
>
> We appreciate the reviewer’s constructive comment. We agree that the magnitude of our relabeling step is heuristic by design; however, this choice was intentional. A small, uniform correction provides a stable and hyperparameter-free refinement step, yielding consistent performance gains across datasets.
>
> | Methods                              | MAE $(\downarrow)$ | CS $(\uparrow)$ |
> | ------------------------------------ | ------------------ | --------------- |
> | No relabeling                        | 4.058              | 73.68           |
> | Different magnitudes (instance-wise) | 4.012              | 72.75           |
> | Proposed (global)                    | **4.002**          | **73.68**       |
>
> To explore more adaptive approaches, we also evaluated instance-specific relabeling (“Different magnitudes”, Table 19, Appendix D.4). Although more flexible in principle, this strategy often produced unstable updates — especially for noisy or difficult samples — and did not outperform the global scheme. These findings indicate that the simple heuristic we employ is both robust and reliable under noisy supervision. We also agree that developing a more principled, data-driven refinement mechanism is a promising direction for future work.
>
> This discussion has been added to Appendix D.4 on pages 19–20 of the revised manuscript.
>
> ***
>
> > **Missing related works**
>
> We thank the reviewer for pointing out these relevant studies. In Section 2 of the revised manuscript, we have cited [1–3]  and clarified their connections to the proposed SOL algorithm. We appreciate the reviewer’s suggestions, which improved both the completeness and clarity of our literature discussion.
>
> Among these works, [2] and [3] focus on noisy-label *classification*, which differs fundamentally from our *ordinal/rank estimation* setting and does not involve ordering constraints.
>
> [1] (DLDL) is more closely related, as it also addresses age estimation and models label ambiguity using a Gaussian kernel. However, DLDL converts each label into a fixed smoothed distribution and optimizes it with KL divergence, without modeling label errors as random variables or addressing ordinal noise. In contrast, SOL models label errors stochastically, imposes ordering constraints in the embedding space, and incorporates an outlier-aware refinement module. Thus, while [1] shares the idea of representing ambiguity probabilistically, our formulation and objectives differ and are tailored specifically to robust rank estimation under noisy supervision.
>
> [1] Gao et al. "Deep label distribution learning with label ambiguity." TIP 2017
>
> [2] Li et al. "Selective-supervised contrastive learning with noisy labels." CVPR 2022
>
> [3] Liu and Guo. "Peer loss functions: Learning from noisy labels without knowing noise rates." ICML 2020
>
> ***
>
> > **Summary**
>
> We have incorporated the reviewer’s suggestions into the revised manuscript, including:
>
> - **comparisons with instance-wise refinements**,
> - **citing and discussing the suggested related works**.
>
> We do appreciate the reviewer’s helpful feedback and would be happy to address any further questions.

---

### Official Review · Reviewer_nqaZ · 2025-10-29

**Soundness:** 3
**Presentation:** 3
**Contribution:** 3
**Rating:** 4
**Confidence:** 3

**Summary:**

The paper proposes an algorithm called Stochastic Order Learning (SOL) for ordinal data rank estimation under label noise. The authors model label errors as random variables and, based on this idea, introduce two loss functions — discriminative loss and stochastic order loss — to learn an embedding space that preserves rank order despite noisy annotations. Additionally, they design an outlier detection and relabeling mechanism based on the learned embeddings to reduce the effect of noisy labels.

**Strengths:**

- The authors correctly identify the limitation of existing ordinal regression and order-learning methods in handling label noise. The stochastic reformulation is an interesting conceptual step forward.
- Combining stochastic modeling of label errors with metric learning objectives is novel within the ordinal regression field.
- The experiments span diverse domains — facial age estimation, aesthetic score prediction, medical image assessment, and text regression — providing broad empirical context.

**Weaknesses:**

- The algorithm repeatedly relies on the exact probability values $𝑝_𝑠$ of the assumed noise distribution in equations (3), (9), and (12)–(17). However, in real-world tasks the noise variance σ is unknown. The authors only fix a constant “test value” 𝜎 test
during inference (see Eq. (22) on page 7 and Appendix C.3). This assumption undermines the theoretical validity of the method: since the noise distribution cannot be known or verified, the resulting stochastic weighting is arbitrary and not grounded in data.
- In equations (8)–(10), the discriminative loss aggregates weighted squared distances between each sample and multiple centroids.
However, this formulation does not ensure that the monotonicity constraint (Eq. (5)) holds.
Appendix A only shows that monotonicity is a sufficient condition, not a necessary one. The authors incorrectly reverse the logic — assuming that minimizing the proposed loss would imply monotonicity.
This is a clear logical inversion error, meaning the optimization process does not guarantee the intended ordered embedding structure.
- All experiments are conducted using synthetic noise (Gaussian, Laplacian, Uniform), despite the paper claiming to handle real-world noisy labels.
The only real-noise experiment (on WMT2020) shows merely about a 2% improvement, which is negligible given the additional model complexity. Although the paper claims the training cost is “acceptable,” Tables 19–21 show that SOL’s training time roughly doubles (87–100% slower) compared to the baseline GOL.
On larger datasets such as RSNA, a single epoch exceeds 1000 seconds — an impractical runtime for real-world use.

**Questions:**

see the Weaknesses

---

> ### Author Response · Authors · 2025-11-17
> **We thank the reviewer for the detailed assessment and constructive critical feedback (Part I).**
>
> Thank you for your constructive review. Please find our responses below; the corresponding revisions have been incorporated into the paper and highlighted in blue.
>
> ***
> > **Unknown $\sigma$ in real-world tasks**
>
> We appreciate the reviewer’s comment. Although the true noise variance is unknown in real-world data, this does not undermine the theoretical validity of SOL. In our formulation, $\sigma$ serves as a modeling parameter that shapes the stochastic ordering function — similar to a bandwidth in density estimation or a temperature parameter in contrastive learning. The theory does not require recovering the true $\sigma$; rather, it characterizes the ordering behavior under a chosen noise model.
>
> In practice, since the true variance is unavailable, we fix $\sigma_\text{test}$ to a default value. This is an engineering choice that determines the smoothness of the stochastic weighting, not an attempt to estimate the actual noise level. Importantly, SOL is not very sensitive to this parameter. As shown in Appendix D.2 and in the WMT2020 results reproduced below, performance remains stable across a wide range of $\sigma_\text{test}$ values.
>
> |$\sigma_\text{test}$| PCC($\uparrow$)|SRCC($\uparrow$)|
> |-|-|-|
> | 0.5 | 0.664 | 0.639|
> | 1.0 (used in the paper) | 0.680 | 0.649|
> | 1.5 | 0.672 | 0.640 |
> | 2.0 | 0.679 | 0.654 |
> | 2.5 | 0.672| 0.656 |
> | 3.0 | 0.670| 0.641 |
> | 3.5 | 0.675| 0.646 |
> | 4.0 | 0.683 | 0.653 |
>
> These results show that SOL’s performance is robust to the choice of $\sigma_\text{test}$, and using a fixed value does not conflict with the theoretical formulation. Please see more analysis on $\sigma_\text{test}$ in Appendix D.2 on pages 18–19.
> ***
>
> > **Monotonicity constraint**
>
> We appreciate the reviewer’s thoughtful analysis. Let us clarify the logical relationship between the following three components:
>
> - **(A) Desideratum (Eq. 4)**
> - **(B) Monotonicity constraint (Eq. 5)**
> - **(C) Discriminative loss (Eqs. 8–10)**
>
> The critique assumes that our method relies on the logical chain **(C) ⇒ (B) ⇒ (A)**, and correctly notes that (C) does not imply the monotonicity constraint (B). However, our formulation does not depend on this chain.
>
> The discriminative loss (C) is not derived from (B) and does not attempt to enforce it. Instead, (C) follows directly from the desideratum (A). As stated in Lines 188–190, the monotonicity constraint (B) is introduced only as a conceptual intuition, illustrating one sufficient geometric condition under which (A) holds.
>
> Formally, if the desideratum (A) is satisfied, each term
>
> $(D_h(x,r_x)-D_h(x,r_x+t)+D_h(x,r_x)-D_h(x,r_x-t))$
>
> in Eq. (8) is non-positive. Minimizing the discriminative loss (C) therefore encourages the model to satisfy (A), without relying on (B).
>
> To avoid misunderstanding, we have clarified in the revised manuscript that the discriminative loss (C) is derived directly from the desideratum (A). Please see page 4.
>
> ***
>
> *(Continued in Part II)*

---

> ### Author Response · Authors · 2025-11-18
> **We thank the reviewer for the detailed assessment and constructive critical feedback (Part II).**
>
> *(Continued from Part I)*
>
> ***
>
> > **Performance improvement on WMT2020**
>
> We disagree that the gain on WMT2020 is negligible. As shown in Table 5, SOL achieves the largest improvement among all methods designed for real-noise settings. RES raises PCC from 0.645 → 0.660 (+0.015), whereas SOL further increases it to 0.680 (+0.020 over RES, +0.035 over Base). SRCC shows a similar pattern (+0.019 over RES). Given the high-entropy human noise in WMT2020, improvements of this size are considered substantial in prior QE literature, indicating that SOL provides a clear robustness benefit. Please see Table 5 and its description on page 9 in the revised paper.
>
> ***
> > **Computational complexity**
>
> The 1160.7 s/epoch time on RSNA does not indicate that SOL is impractical. This cost reflects training only — not inference — and is comparable to strong ranking baselines such as MWR. For reference, the table below reports the per-epoch training times on RSNA. We also emphasize that training is performed once offline, while real-world deployment is dominated by inference.
>
> | Algorithm | Training time per epoch|
> |-|-|
> | MWR | 1036.3s |
> | GOL | 664.1s |
> | SOL | 1160.7s |
>
> Importantly, SOL adds no inference overhead. It is even faster than GOL at test time (0.091 s vs. 0.123 s), since the costly pairwise probability terms are computed only during training.
>
> | Algorithm | Feature extraction | Inference | Total |
> |-|-|-|-|
> | GOL | 0.040s | 0.083s | 0.123s |
> | SOL | 0.040s | 0.051s | 0.091s |
>
> The larger epoch cost on RSNA stems from serial data loading (num_workers = 1 for comparability with prior work), not from SOL itself. Standard data-loading parallelization reduces the time by over 5$\times$, to about 220 s/epoch:
>
> | num_workers | Training time per epoch |
> |-|-|
> |1|1160.7s|
> |8|223.6s|
>
> Thus, the previously reported 1160.7 s/epoch is a conservative upper bound caused by I/O bottlenecks. With parallel data pipelines, SOL trains efficiently and remains fully practical for real-world deployment.
>
> This complexity issue has been discussed in detail in Appendix D.6 on pages 21-22.
>
> ***
> > **Summary**
>
> We have addressed all of the reviewer’s concerns in the revised manuscript, including:
>
> - **role of $\sigma$ in the theoretical model**,
> - **relationship among the desideratum, monotonicity constraint, and discriminative loss**,
> - **detailed analysis of training and inference complexity**.
>
> We appreciate the reviewer’s thoughtful feedback and would be happy to address any further questions.

---

### Official Review · Reviewer_Bpfh · 2025-10-29

**Soundness:** 4
**Presentation:** 4
**Contribution:** 3
**Rating:** 8
**Confidence:** 3

**Summary:**

This paper proposes a novel method, Stochastic Order Learning (SOL), for robust rank estimation on label-noisy data. The method models label errors as random variables and learns an embedding space where each instance is encouraged to approach its stochastically related rank centroids. To achieve this, the authors design two loss functions, the discriminative loss and the stochastic order loss. After training, the method further improves data quality by detecting and relabeling outliers (instances with extreme label errors). Experiments on various datasets (facial age estimation, aesthetic score regression, medical imaging, and textual regression) demonstrate high accuracy and strong robustness to label noise.

**Strengths:**

his paper's strengths are as follows.

(1) This paper is the first study to address rank estimation with label noise, a setting that is pervasive in real-world scenarios, as noted in the paper. The contribution is thus highly significant for practical applications.

(2) The paper introduces a natural probabilistic model of label errors for rank estimation and proposes an appropriate learning framework based on this probabilistic formulation.

(3) The method demonstrates robust performance under both artificially generated and naturally occurring label noise.

**Weaknesses:**

This paper's weakness is as a follow.

(1)  The paper assumes label noise as formulated in Equation (2), but it remains unclear how the noise parameter σ is determined in real-world problems. During training, σ controls the amount of label corruption and is therefore crucial, yet in practice, this parameter is typically unknown. How is σ selected or estimated in real-world settings?

**Questions:**

(1) In real-world scenarios, does label noise actually follow the distribution assumed in Equation (2)? Moreover, how do the authors verify that such a type of label noise occurs in real data?

(2) From the quantitative results in Appendix D.3, the impact of outlier detection and relabeling appears very small. Why does removing outliers not lead to a more noticeable quantitative improvement?

(3) It is recommended to include visualizations of outlier detection on real-noise datasets (e.g., WMT2020).  Since detecting real noisy labels would be highly beneficial in practice, this would better showcase the potential of the proposed approach.

---

> ### Author Response · Authors · 2025-11-17
> **We are grateful for the reviewer’s strong endorsement and helpful observations.**
>
> Thank you for your positive review and insightful suggestions. Please find our responses below; the corresponding revisions have been incorporated into the paper and highlighted in blue.
> ***
> > **Label noise in real-world**
>
> We appreciate the reviewer’s thoughtful question. Many rank-estimation datasets — including CLAP2015, AADB, and RSNA — obtain ground-truth labels by averaging multiple independent annotators’ scores. Such averaged labels are well known to approximate a Gaussian distribution due to the central-limit effect, and CLAP2015 even provides per-sample variances reflecting this assumption. While individual annotator noise may deviate from Gaussian, the aggregated labels are empirically close to Gaussian, making the formulation in Eq. (2) a reasonable and widely adopted modeling choice.
>
> Regarding the choice of $\sigma_\text{test}$ in the real-world WMT2020 dataset, we follow the same practice across all datasets by fixing it to 1.0. Importantly, SOL is not highly sensitive to this parameter; preliminary sweeps consistently showed stable performance across a broad range of $\sigma_\text{test}$ values. Thus, $\sigma_\text{test}=1.0$ was selected as a robust default rather than a finely tuned hyperparameter. To demonstrate this insensitivity, we report the performance on WMT2020 under different $\sigma_\text{test}$ values below.
>
> |$\sigma_\text{test}$| PCC($\uparrow$)|SRCC($\uparrow$)|
> |-|-|-|
> |0.5|0.664|0.639|
> |1.0|0.680|0.649|
> |1.5| 0.672|0.640|
> |2.0|0.679|0.654|
> |2.5|0.672|0.656|
> |3.0|0.670|0.641|
> |3.5|0.675|0.646|
> |4.0|0.683|0.653|
>
> The complete discussion of real-world label noise has been incorporated into Appendix D.2 on pages 18–19.
> ***
>
> > **Impact of outlier relabeling**
>
> We understand the reviewer’s point that the quantitative gain from the relabeling module appears small. This is expected;  datasets such as CLAP2015 are relatively small, so the absolute number of severe outliers is limited, and only a few samples are actually corrected.
>
> On larger datasets — such as RSNA — the effect becomes clearer because more outliers are detected, making the refinement step more influential. The table below shows this trend on RSNA. These results have been discussed in Appendix D.4 on page 19.
>
> ||Gaussian ($\kappa=0.1$)||Gaussian ($\kappa=0.15$)||Gaussian($\kappa=0.2$)||
> |-|-|-|-|-|-|-|
> |Algorithm|MAE($\downarrow$)|CS($\uparrow$)|MAE($\downarrow$)|CS($\uparrow$)|MAE($\downarrow$)|CS($\uparrow$)|
> | w/o label refinement |7.967|**81.50**|7.800|79.50|8.196|74.00|
> | w/ label refinement |**7.579**|78.50|**7.706**|**80.50**|**8.051**|**76.50**|
>
> ***
> > **Visualizing outliers on real-noise dataset**
>
> We appreciate the reviewer’s constructive suggestion. Following your recommendation, we performed an additional qualitative analysis of outlier detection on the real-noise WMT2020 dataset, where annotation noise arises naturally from diverse translation-quality judgments. Unlike synthetic noise, these discrepancies reflect genuine human variability — e.g., harsh ratings for fluent translations or overly generous ratings for mistranslated outputs.
>
> To illustrate how SOL identifies such inconsistencies, we categorize representative cases into two types.
>
> - Type A: good translations receiving abnormally low human scores,
> - Type B: poor translations receiving surprisingly high scores.
>
> The examples below show clear cases where annotated scores deviate substantially from both linguistic quality and the model’s predictions.
>
> | Type | Real Score | Pred Score | Source Text | Translation | Issue |
> |------|------------|------------|-------------|-------------|--------|
> | A1 | 4 | 22 | Не по человеку спесь. | Don’t rush into it. | Fluent sentence but unusually low human score. |
> | A2 | 6 | 17 | Не пеняй на зеркало , коль рожа крива. | Don’t foam at the mirror if it’s crooked. | Acceptable fluency, score is unrealistically low. |
> | B1 | 66 | 6 | Задком, кувырком, да и под горку. | Backward, somersault, and downhill. | Literal mistranslation; idiomatic meaning (“things going downhill”) is lost. |
> | B2 | 56 | 8 | Религия яд – береги ребят | Religion Poison – Save the Children | Ungrammatical; missing verb (“Religion is poison”), resulting in awkward phrasing. |
> | B3 | 67 | 15 | Что за чудак, да и чудило. | What a freak, and a miracle. | Semantic error; “чудило” mistranslated as “miracle,” losing intended meaning. |
>
> These results confirm that SOL effectively identifies inconsistent annotations in real-world translation datasets. The full analysis has been added to Appendix D.9 on page 24.
>
> ***
> > **Summary**
>
> We have incorporated all of the reviewer’s suggestions into the revised manuscript, including:
>
> - **clarification of real-world label-noise modeling**,
> - **expanded experiments on outlier relabeling**,
> - **qualitative visualization of outliers on WMT2020**.
>
> We appreciate the reviewer’s positive assessment and would be happy to clarify any additional points.

---

### Official Review · Reviewer_8PLT · 2025-10-31

**Soundness:** 3
**Presentation:** 3
**Contribution:** 3
**Rating:** 6
**Confidence:** 3

**Summary:**

This paper proposes an algorithm called Stochastic Order Learning (SOL) for robust and reliable rank estimation in the presence of label noise. The key idea is to model label errors as random variables, following a discrete Gaussian distribution, and to learn an embedding space where instances are arranged according to their true ranks despite noisy labels. The method introduces two loss functions—discriminative loss and stochastic order loss, which are to enforce geometric constraints in the embedding space. Additionally, SOL includes an outlier detection and relabeling mechanism to refine the training data. Extensive experiments on facial age estimation, aesthetic score regression, medical assessment, and textual regression datasets demonstrate that SOL outperforms existing noise-robust classification, regression, and rank estimation methods under various synthetic and real-world noise settings.

**Strengths:**

1.	The paper tackles an important problem - label noise in ordinal regression.
2.	Extensive experiments across multiple domains (computer vision and natural language processing) and various noise types (Gaussian, Laplacian, and Uniform) demonstrate the effectiveness of the proposed approach.

**Weaknesses:**

1.	The method's performance is tied to the assumption of a symmetric, unimodal noise distribution, which is a key limitation in practical applications.
2.	Ablation studies and hyperparameter analysis are heavily focused on CLAP2015. It is unclear if the same settings and component importance hold for datasets with the largest gains (e.g., GDELT is not mentioned in this context, but the principle applies to datasets where SOL shines).
3.	The method introduces non-trivial computational cost compared to non-stochastic baselines, which could be a constraint.

**Questions:**

1.	The method assumes a symmetric noise model (Eq. 2). How would SOL perform if the real-world label noise is asymmetric (e.g., annotators consistently over-estimate ages)?
2.	The hyperparameter σtest is fixed during inference. Could the performance be further improved by making it adaptive or by estimating it from the data?
3.	The outlier relabeling uses a global average correction (Eq. 20). Have you explored instance-specific relabeling strategies, and why was a uniform correction chosen?
4.	The computational cost is higher than GOL. Are there strategies to improve the efficiency of the stochastic distance computations?

---

> ### Author Response · Authors · 2025-11-17
> **We thank the reviewer for the encouraging assessment and constructive comments.**
>
> Thank you for your positive review and insightful suggestions. Please find our responses below; the corresponding revisions have been incorporated into the paper and highlighted in blue.
> ***
> > **Asymmetric noise**
>
> To evaluate SOL under asymmetric noise, we added skewed perturbations sampled from $\text{SkewNorm}(a=5,\mu=0,\sigma=\kappa\sigma_{\cal X})$. Results are shown below. All methods degrade under skewed noise, but SOL remains the best performer on every dataset, confirming the robustness of its stochastic modeling. We sincerely appreciate the reviewer’s suggestion, which strengthens the validation of SOL under broader noise conditions.
>
> | | MORPH ||CLAP2015 ||AADB ||RSNA ||
> |-|-|-|-|-|-|-|-|-|
> | Algorithm  | MAE$(\downarrow)$ | CS$(\uparrow)$ | MAE$(\downarrow)$ | CS$(\uparrow)$ |MAE$(\downarrow)$ | CS$(\uparrow)$ |MAE$(\downarrow)$ | CS$(\uparrow)$ |
> | GOL | 3.351 | 82.51 | 4.407| 68.40| 0.120 | 91.00|8.994|71.00|
> | SOL (Proposed) | **3.296**  | **83.15** |**4.379**|**69.97**| **0.118**|**92.30**|**8.544**|**73.00**|
>
> These results have been incorporated into Tables 1–4 of the revised paper and will continue to be updated.
>
> ***
> > **Adaptive $\sigma_\text{test}$**
>
> To investigate whether $\sigma$ can be estimated from data, we added a lightweight head that predicts the mean $\mu$ and standard deviation $\sigma$, trained with a Gaussian NLL loss so that the predicted $\sigma$ replaces the constant in Eq. (2). We evaluated two variants:
> (1) *Joint training*, where the $\sigma$-prediction head and SOL are optimized together, and
> (2) *Two-stage scheme*, where the $\sigma$-prediction head is trained first and then frozen during SOL training.
> In practice, both approaches showed instability due to noisy labels affecting $\sigma$-prediction, and both underperformed the fixed-$\sigma$ baseline. As shown below for CLAP2015 at $\kappa = 0.4$, the fixed setting provides better MAE and CS.
> |Method |MAE($\downarrow$)|CS($\uparrow$)|
> |-|-|-|
> |Joint adaptive $\sigma_\text{test}$|5.032|67.10|
> |Two-stage adaptive $\sigma_\text{test}$|4.171|71.64|
> |Fixed $\sigma_\text{test}$|**4.002** |**73.68**|
>
> These results have been incorporated into Table 15 on page 19 of the revised manuscript.
> ***
> > **Choice of relabeling strategy**
>
> We explored an instance-specific relabeling strategy in Table 19 on page 20, denoted as “Different magnitudes.” The results are reproduced below.
> |Methods | MAE $(\downarrow)$ | CS $(\uparrow)$ |
> |-|-|-|
> |No relabeling|4.058| 73.68 |
> |Different magnitudes| 4.012 | 72.75|
> |Proposed|**4.002** | **73.68** |
>
> Although instance-specific relabeling may seem more flexible, it depends on per-sample discrepancies between noisy and estimated ranks. Outliers have the least reliable estimates, making these discrepancies highly variable and prone to unstable or overly aggressive corrections. In contrast, our uniform correction applies a small, consistent adjustment to detected outliers, avoiding noisy sample-specific shifts. As a result, the uniform strategy yields the most stable and reliable gains.
> ***
> > **Computational cost**
>
> SOL can be further accelerated through two lightweight strategies, evaluated on CLAP2015 ($\kappa=0.4$). First, *centroid subsampling* reduces distance evaluations with minimal accuracy loss.
> | Sampling ratio | MAE| Training time per epoch (s) |
> |-|-|-|
> | 0.1 |4.029| 47.9s |
> | 0.2 |4.018| 48.2s |
> | 1.0 |4.002| 52.1s |
>
> Second, *FP16 distance computation*  lowers runtime while preserving accuracy.
> |Precision|MAE|Training time per epoch|
> |-|-|-|
> |FP16|4.008|48.0s|
> |FP32|4.002|52.1s|
> These efficiency options have been discussed in Appendix D.6 on pages 21–22.
> ***
> > **Ablation studies and analysis on other datasets**
>
> To confirm that our design choices generalize beyond CLAP2015, we conducted the same ablation studies on RSNA (under Gaussian noise with $\kappa=0.15$) and WMT2020 — two datasets where SOL shows substantial gains.
> |Method|$l_{\text{disc}}$|$l_{\text{order}}$| RSNA||WMT2020||
> |-|-|-|-|-|-|-|
> ||||MAE($\downarrow$)|CS($\uparrow$)|PCC($\uparrow$)|SRCC($\uparrow$)|
> |I|$\checkmark$||88.357|4.50|0.396|0.354|
> |II||$\checkmark$|8.040|77.50|0.673|0.634|
> |III|$\checkmark$|$\checkmark$|**7.706**|**80.50**|**0.680**|**0.649**|
>
> The same pattern holds across both datasets; the combination of $l_{\text{disc}}$ and $l_{\text{order}}$ yields the best performance. These results have been added in Appendix D.10 on page 24.
> ***
> > **Summary**
>
> We have incorporated all suggested analyses into the revision, including:
> - **evaluations under asymmetric noise**,
> - **experiments with adaptive** $\sigma_\text{test}$,
> - **comparisons of relabeling strategies**,
> - **computational efficiency improvements**,
> - **ablation studies on RSNA and WMT2020**.
>
> We appreciate the reviewer’s positive assessment and would be happy to provide any additional clarification.

---

> ### Author Response · Authors · 2025-12-01
> **Updates to the Asymmetric Noise Evaluation**
>
> All updates related to the asymmetric noise evaluation have now been finalized and incorporated into the revised manuscript. We genuinely appreciate the constructive feedback, which helped strengthen our work.

---

### Official Review · Reviewer_BwZ6 · 2025-11-02

**Soundness:** 2
**Presentation:** 3
**Contribution:** 2
**Rating:** 2
**Confidence:** 4

**Summary:**

The proposed stochastic order learning method frames orders as random variables, and develops discriminative loss and stochastic order loss to optimize network parameters. Experiments are conducted on benchmark facial age estimation datasets. Results show its superiority over baselines under different noise distributions.

**Strengths:**

-The proposed method models label errors as random variables and provides a solid theoretical basis.

-Stochastic order learning method is not sensitive to the prior noise distribution, shown in Table 1~4. Different noise distributions, such as Gaussian, Laplacian and uniform distribution, lead to similar performance.

-Extensive experiments are conducted on various datasets and results show its effectiveness for the age estimation task.

**Weaknesses:**

-Baseline methods are not comprehensive. A naïve method is to utilize these mature ranking loss functions in Learning to Rank methods, like RankNet and SoftRank. Similar idea has been implemented in SoftRank. These kinds of methods should be compared in the experiments.

-Compared with GOL in Table 1~4, the performance improvement of stochastic order learning method is marginal.

-Compared with those benchmark loss functions, the computation complexity of the proposed stochastic order loss is higher. Moreover, the time complexity of the proposed loss function should be provided.

**Questions:**

Ranking loss functions, like RankNet and SoftRank, are not compared in the experiments.

---

> ### Author Response · Authors · 2025-11-17
> **We thank the reviewer for the constructive and insightful review.**
>
> Thank you for the constructive feedback. We address the raised points below; the corresponding revisions have been incorporated into the paper and highlighted in blue.
>
> ***
> > **Comparison to *Learning-to-Rank* methods**
>
> We appreciate the suggestion to compare with RankNet and SoftRank. These methods, however, are designed for information-retrieval scenarios where the goal is relative ordering rather than predicting absolute ordinal values. Because any monotonic transformation of the predicted scores preserves the ranking (for example, 10-20-30 and 1-2-3 yield the same ordering), their outputs are not calibrated and cannot be directly evaluated with absolute metrics, such as MAE or CS, nor used to estimate quantities like age.
>
> To enable a fair comparison, we implemented RankNet and SoftRank in our experimental setting. Both models were trained with the same VGG16 backbone, and we followed standard practice by applying k-NN regression in their embedding spaces to convert relative scores into absolute rank estimates. As shown in the table below, SOL consistently outperforms both baselines on MORPH II across all noise types. These results have been added to Appendix D.8 in page 23.
>
> ||Gaussian $\kappa=0.2$||Gaussian $\kappa=0.3$||Gaussian $\kappa=0.4$||Laplacian $\kappa=0.3$||Uniform $\kappa=0.3$||
> |-|-|-|-|-|-|-|-|-|-|-|
> | Algorithm | MAE$(\downarrow)$ | CS$(\uparrow)$ | MAE$(\downarrow)$ | CS$(\uparrow)$ | MAE$(\downarrow)$ | CS$(\uparrow)$ | MAE$(\downarrow)$ | CS$(\uparrow)$ |MAE$(\downarrow)$ | CS$(\uparrow)$ |
> | RankNet | 2.639 | 89.80 | 2.990 | 86.16 | 3.116 | 82.79 | 3.146 | 84.15 | 2.634 | 88.89 |
> | SoftRank($\sigma_s=1.0$)| 3.147 | 83.06 | 3.394 | 81.97 | 3.427 | 80.15 | 3.801 | 75.96 | 3.099 |85.34|
> | SOL |**2.489**|**91.35**|**2.663**|**89.62**|**2.826**|**87.70**|**2.986**|**85.88**| **2.499**|**90.89**|
> &nbsp;
>
> ***
>
> > **Comparison to GOL**
>
> While the improvement over GOL may appear small on cleaner datasets such as MORPH II — where label noise is limited and the MAE is already near its lower bound — the advantage of SOL becomes more evident in challenging high-noise environments. On CLAP2015, for example, the MAE gap increases from $0.065$ at $\kappa  = 0.2$ to $0.103$ at $\kappa = 0.4$, indicating that SOL’s robustness strengthens as noise grows.
>
> A similar trend is observed on RSNA (more than +1.0 CS) and on WMT2020 with real subjective noise (+2.0 PCC and +1.9 SRCC). Small differences on easy datasets therefore translate into clear performance gains under realistic noisy settings. In short, the stochastic modeling in SOL provides reliable benefits precisely where deterministic order-learning methods degrade. We have clarified this point in L456–458 on page 9 of the revised manuscript.
>
> ***
>
> > **Computational complexity**
>
> To address the concern regarding computational cost, we compared SOL with the ranking baselines suggested by the reviewer (RankNet and SoftRank), as well as recent state-of-the-art rank-estimation methods (MWR and GOL). All experiments were conducted on CLAP2015 using an RTX 4090 GPU. The table below reports the per-epoch training times.
>
> SoftRank is computationally heavy due to its recursive rank-binomial construction, and MWR also requires expensive, iterative window-based rank estimation. SOL, by contrast, provides stochastic-order modeling in a simpler and more efficient manner. As the table shows, SOL is much faster than SoftRank and MWR, and although it is slower than GOL, the overhead is modest and does not present a practical training limitation.
>
> | Algorithm | Training time per epoch|
> |-|-|
> | RankNet | 44.8s |
> | SoftRank | 96.2s |
> | MWR | 77.3s |
> | GOL | 27.8s |
> | SOL | 52.1s |
>
> We also compared GPU memory consumption during loss computation (batch size = 32):
>
> | Algorithm | Memory |
> |-|-|
> | GOL | 8.19 MB |
> | SOL | 0.60 MB |
>
> GOL requires significantly more memory because it constructs dense pairwise direction tensors and expanded index structures, leading to large intermediate buffers. SOL avoids these operations and computes pairwise probabilities on the fly, resulting in a much smaller memory footprint.
>
> Overall, SOL provides a favorable balance between computational efficiency and modeling power, and is practical for real-world use. Additional complexity analyses have been added to Appendix D.6 on page 21 of the revised manuscript.
>
> ***
>
> > **Summary**
>
> We have incorporated all suggested analyses into the revision, including:
>
> - **LTR comparisons** (RankNet, SoftRank),
> - **robustness advantages over GOL**,
> - **expanded complexity evaluation**.
>
> We hope these updates address your concerns, and we would be happy to provide any additional clarification.

---

### Author Response · Authors · 2025-11-12

We would like to thank all reviewers for their time and effort in providing valuable feedback. We will upload our responses to each question or comment as soon as possible.

---

### Author Response · Authors · 2025-12-01
**Completion of Updates in the Revised Manuscript**

We would like to note that all updates—previously indicated as ongoing—have now been fully incorporated into Tables 1–4 in the revised manuscript. We would be grateful if the Area Chair and reviewers could refer to the updated version.

---

### Note · Authors · 2026-01-27

I have read and agree with the venue's withdrawal policy on behalf of myself and my co-authors.

---

### Meta-Review · Area_Chair_cdDc · 2026-01-11

**Summary:**

The paper addresses the problem of ordinal regression. The Authors claim that they introduce a robust and reliable rank estimation method in the presence of label noise. Their approach models label errors as random variables using a discrete Gaussian distribution and learns an embedding space in which instances are stochastically ordered according to their true ranks. The method further refines the data by detecting and relabeling outliers in the learned embedding space. Extensive experiments are conducted on facial age estimation, aesthetic score regression, medical assessment, and textual regression datasets.

The paper received highly divergent scores, ranging from 2 to 8. The main concerns include the absence of discussion or comparison with learning-to-rank methods, limited experimental evaluation and marginal empirical improvements, unclear computational complexity, the restrictive assumption of symmetric unimodal noise, an unclear procedure for estimating noise parameters, and potential flaws in the theoretical derivations. The Authors have partially addressed some of these critical remarks in their rebuttal.

As Area Chair, I examined the paper more closely and share the Reviewers’ view that the noise model and the experimental methodology constitute the main weaknesses. Most notably, neither the Authors nor the Reviewers reference classical noise models for ordinal regression, such as the grouped continuous model or the stereotype model (e.g., *Regression Models for Ordinal Data*, JSTOR 1980; *Regression and Ordered Categorical Variables*, JSTOR 1984). Even if the proposed model offers advantages over these classical approaches, a proper discussion is expected in a submission to a top-tier conference. It is also worth noting that related ideas such as outlier detection and data monotonization have been studied previously (e.g., *Isotonic Separation*, INFORMS Journal on Computing; *Rule Learning with Monotonicity Constraints*, ICML 2009; Isotonic Classification), and their relation to the present work is not discussed.

**Reviewer Concerns:**

The Authors have been able to address the critical remarks at least in part. However, the noise model and the experimental methodology remain the main weaknesses of the paper. Most importantly, the paper lacks any discussion or references to classical ordinal regression models. These issues would likely be raised during the discussion phase and cannot be easily remedied before its conclusion.

**Reviewer Scores:**

Most likely all Reviewers who gave scores 6 and higher would lower their evaluation and agree that the paper in the current form is not yet ready for publication.

---

### Decision · Program_Chairs · 2026-01-26

Reject